# Core Knowledge Deficits in Multi-Modal Language Models

Yijiang Li[1]  Qingying Gao[*2]  Tianwei Zhao[*2]  Bingyang Wang[*3]  Haoran Sun[2]  Haiyun Lyu[4]
Robert D. Hawkins[5]  Nuno Vasconcelos[1]  Tal Golan[6]  Dezhi Luo[78]  Hokin Deng[9]

## Abstract

While Multi-modal Large Language Models (MLLMs) demonstrate impressive abilities over high-level perception and reasoning, their robustness in the wild remains limited, often falling short on tasks that are intuitive and effortless for humans. We examine the hypothesis that these deficiencies stem from the absence of core knowledge—rudimentary cognitive abilities innate to humans from early childhood. To explore the core knowledge representation in MLLMs, we introduce **CoreCognition**, a large-scale benchmark encompassing 12 core knowledge concepts grounded in developmental cognitive science. We evaluate 230 models with 11 different prompts, leading to a total of 2,530 data points for analysis. Our experiments uncover four key findings, collectively demonstrating core knowledge deficits in MLLMs: they consistently underperform and show reduced, or even absent, scalability on low-level abilities relative to high-level ones. Finally, we propose *Concept Hacking*, a novel controlled evaluation method, that reveals MLLMs fail to progress toward genuine core knowledge understanding, but instead rely on shortcut learning as they scale. Project page at https://williamium3000.github.io/core-knowledge/.

## 1. Introduction

Are human minds born with knowledge (Plato et al., 1763)? This has been the central question of Western thoughts since the ancient Greeks (Russell, 1946). Socrates and Plato both believe that humans must be born with a set of innate knowledge. In *Meno, 80d–86b*, Socrates introduces the theory of *anamnesis* (recollection), where he suggests our "soul is immortal", and "it can recollect the things it knew before" (Fowler et al., 1914). Plato further sets the distinction between innate knowledge and those we gain through experience: in *Republic VII*, the Allegory of the Cave, he suggests that our experiences are *skiés*, like shadows on the cave wall, which are contingent instantiations of the *eidos*, the knowledge born with our minds. One example of *eidos* is our understanding of a circle: while a perfect circle never exists in reality, we still understand what it means to be a perfect circle (Jowett et al., 1888). Kant's view is more intricate: he suggests we never have an innate knowledge of *noumena*, "things-in-themselves", but we have knowledge of *phenomenon*, "things-about-themselves", meaning we only are born with knowledge about the structures of our experiences, such as causality, permanence, and continuity, but never gifted with knowledge of experiences in itself (Kant, 1781). In other words, we have innate, core knowledge about basic domains of the world.

We are closer than ever to achieving human-level artificial intelligence. By training on vast web-scale corpora and scaling to hundreds of billions of parameters, Large Language Models (LLMs) now surpass expert humans in knowledge- and reasoning-intensive tasks (Brown et al., 2020; Achiam et al., 2023; Bai et al., 2023a; Touvron et al., 2023; Jaech et al., 2024). These capabilities extend beyond language: with modality alignment (Liu et al., 2024b; Li et al., 2023b; Zhu et al., 2023), MLLMs exhibit unprecedented high-level perception and reasoning (Gemini, 2023; Wu & Xie, 2024; Xu et al., 2024; Yang et al., 2025a; Shao et al., 2024; Yang et al., 2024; Li et al., 2024a; Fu et al., 2023), mastering tasks such as chart understanding (Masry et al., 2022), geometry and math (Lu et al., 2023), and action recognition and prediction (Ying et al., 2024; Liu et al., 2024c), often reaching or exceeding human performance (Huang & Zhang, 2024).

Despite advances in high-level perception and reasoning abilities, state-of-the-art MLLMs still underperform humans on simple and rudimentary tasks such as counting (Paiss et al., 2023; Qharabagh et al., 2024), perspective taking (Tang et al., 2025b), spatial reasoning (Zhang et al., 2025; Tang et al., 2025a), temporal reasoning(Saxena et al., 2025),

*Equal contribution [1]University of California San Diego [2]Johns Hopkins University [3]Emory University [4]University of North Carolina at Chapel Hill [5]Stanford University [6]Ben-Gurion University of the Negev [7]University of Michigan [8]University College London [9]Carnegie Mellon University. Correspondence to: Yijiang Li <yijiangli@ucsd.edu>, Dezhi Luo <ihzedoul@umich.edu>, Hokin Deng <hokind@andrew.cmu.edu>.

*Proceedings of the 42nd International Conference on Machine Learning*, Vancouver, Canada, PMLR 267, 2025. Copyright 2025 by the author(s).

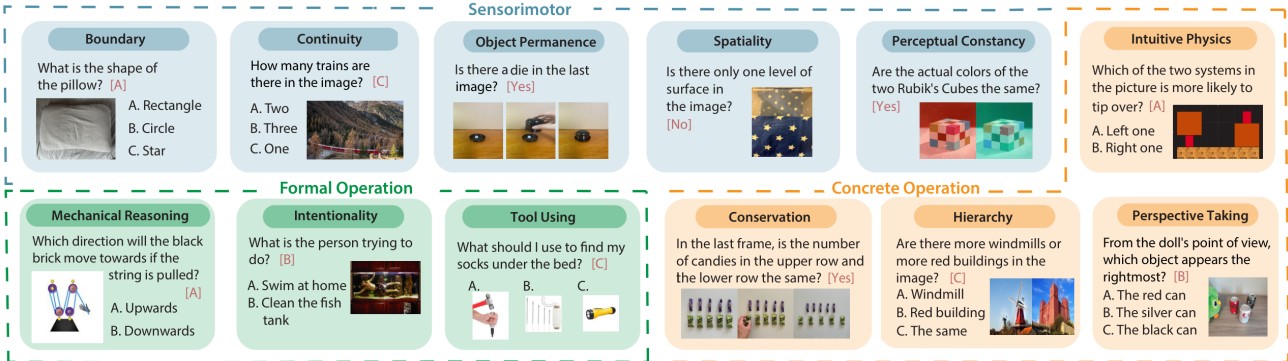

Figure 1. Examples from our **CoreCognition** benchmark.

| Concept | Definition | Concept | Definition | Concept | Definition |
|---|---|---|---|---|---|
| **Boundary** | The transition from one object to another. | **Continuity** | Objects persist as unified, cohesive entities across space and time. | **Permanence** | Objects do not cease to exist when they are no longer perceived. |
| **Spatiality** | The *a priori* understanding of the Euclidean properties of the world. | **Perceptual Constancy** | Changes in appearances don't mean changes in physical properties. | **Intuitive Physics** | Intuitions about the laws of how things interact in the physical world. |
| **Perspective** | To see what others see. | **Hierarchy** | Understanding of inclusion and exclusion of objects and categories. | **Conservation** | Invariances of properties despite transformations. |
| **Tool Use** | The capacity to manipulate specific objects to achieve goals. | **Intentionality** | To see what others want. | **Mechanical Reasoning** | Inferring actions from system states and vice versa. |

Table 1. Abbreviated definitions of the 12 core abilities assessed. See Appendix A.2 for details.

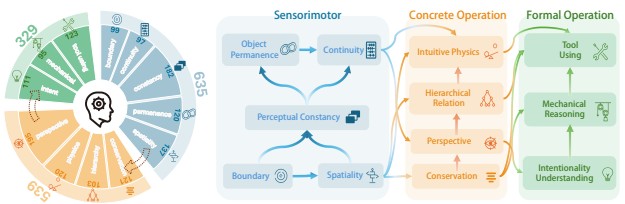

Figure 2. **Left.** Statistics of the **CoreCognition** benchmark. **Right.** Construction of taxonomy. Dependencies between abilities are indicated with arrows.

and compositional reasoning (Yuksekgonul et al., 2022; Sahin et al., 2024; Mitra et al., 2024)—tasks that are intuitive and effortless for humans, even as MLLMs excel at related high-level reasoning (Paiss et al., 2023; Rahman-zadehgervi et al., 2024), exemplifying the long-standing Moravec's Paradox (Moravec, 1988). This high-level excellence often fails to generalize to out-of-distribution or real-world scenarios, where small changes in task conditions can cause significant performance drops (Shiffrin & Mitchell, 2023; Zhang et al., 2024b; Bai et al., 2024; Oh et al., 2025; Dong et al., 2025). Moreover, MLLMs are vulnerable to imperceptible perturbations (Schlarmann & Hein, 2023), susceptible to prompt variations (Wu et al., 2023), and can be easily jailbroken to generate unsafe or unregulated content (Wang et al., 2024b; Gu et al., 2024; Li et al., 2024c).

In this work, we hypothesize that the deficiencies observed in MLLMs stem from the absence of core knowledge—fundamental cognitive abilities innately present in humans from early childhood that underpin advanced reasoning. To examine this hypothesis, we explore the existence, representation, and use of core knowledge in MLLMs by introducing the first large-scale benchmark tailored for core knowledge—**CoreCognition**. It comprises 1,503 samples with over 95 samples for each concept, as exemplified in Fig. 1. Drawing on insights from developmental cognitive science, we propose a taxonomy of 12 abilities encompassing the full spectrum of core knowledge, from basic cognitive skills to advanced reasoning.

To provide a comprehensive evaluation of core knowledge over the existing MLLMs, we assess a total of 230 models with 11 different prompting techniques, yielding a total of 2,530 data points. Leveraging these results, we analyze model performance across varying levels of core ability, examining the inter-dependencies among core knowledge and their predictive power for higher-level reasoning and perception, as well as the scaling effect (performance across different model sizes) To further ascertain core knowledge deficits in MLLMs, we design controlled experiments that manipulate causal features within images to perturb the ground-truth labels, allowing us to determine whether models genuinely possess the targeted core knowledge or merely approximate it through shortcuts and spurious correlations. Our key findings are:

- Core Knowledge Deficits: MLLMs consistently perform worse on low-level abilities compared to high-level abilities (Sec. 4.1).

- Misaligned Dependency: MLLM performance on high-level abilities is not correlated with the underlying low-level abilities that support them (Sec. 4.2).
- Not Scaling: MLLMs exhibit less, or even no, scalability (with respect to increasing model parameters) on low-level abilities compared to high-level abilities. (Sec. 4.4).
- Models increasing in size exhibit core deficits and shortcut-taking behaviors rather than progressing toward conceptual understanding. (Sec. 5.2).

## 2. Related Works

**Multi-modal Large Language Models.** With the advent of large language models (LLMs), state-of-the-art (SOTA) MLLMs (Liu et al., 2024b; Li et al., 2023c) have adopted open-source LLMs (Touvron et al., 2023; Peng et al., 2023; Jiang et al., 2023) and aligned visual features to the LLM embedding space (Li et al., 2023b). To enable open-ended conversational abilities, LLaVA (Liu et al., 2024b) distills ChatGPT's conversational skills into MLLMs, resulting in substantial performance gains—a process that has become standard practice in the field (Wang et al., 2023; Bai et al., 2023a; Gemini, 2023; Team, 2024a; Sun et al., 2023; Li et al., 2022a).

**Benchmarks for Multi-modal Large Language Model.** A wide range of benchmarks have been proposed to evaluate the growing capabilities of MLLMs, ranging from vision question answering (VQA) (Antol et al., 2015; Marino et al., 2019), image captioning (Plummer et al., 2015; Lin et al., 2014), OCR and text understanding (Liu et al., 2023b). More recently, MLLM benchmarks have focused on higher-level reasoning, such as MathVerse (Zhang et al., 2024a) and ScienceQA (Lu et al., 2022), emphasizing multimodal reasoning in scientific domains. Of particular relevance are M3GIA (Song et al., 2024) and Marvel (Jiang et al., 2024), which address cognitive complexity, abstraction, and multi-step reasoning, but primarily focus on task coverage or high-level general intelligence. In contrast, **CoreCognition** targets early-emerging core abilities that support higher-level perception and reasoning. DevBench (Tan et al., 2024) adopts a developmentally inspired framework to probe language learning trajectories, but focuses solely on language, unlike our benchmark, which targets multi-modal core knowledge.

**Shortcut Learning.** Shortcut learning is closely related, especially to *Concept Hacking*. Deep learning models are prone to exploiting spurious correlations—features that enable strong in-distribution performance but lead to brittleness out of distribution (Alvi et al., 2018). Early work addressed this by removing biased features from learned embeddings (Wang et al., 2019). Subsequent approaches trained auxiliary bias predictors to identify shortcut cues and encouraged the main model to predict against them (Bahng et al., 2020; Cadene et al., 2020; Nam et al., 2020;

Clark et al., 2019; Dagaev et al., 2021). Other methods reduce shortcut reliance by minimizing mutual information between features and bias attributes (Kim et al., 2019), or by generating bias-conflicting samples through latent factor swapping (Lee et al., 2021).

**Core Knowledge in Humans.** The debate over core knowledge has historically framed nativist and empiricist epistemologies (Plato et al., 1763; Kant, 1781; Russell, 1946), and since the cognitive revolution, has shifted toward empirical investigation (Piaget, 1950; Fodor, 1975). Piaget's stage-based theory and subsequent research established the foundations of developmental psychology (Piaget & Inhelder, 1969; Barrouillet, 2015; Spelke et al., 1992; Rochat, 2024; Carey et al., 2015). Recent advances show that even infants exhibit rudimentary knowledge of objects (Baillargeon & Carey, 2012; Kar et al., 2019; Ullman & Tenenbaum, 2020), actions (Yang et al., 2015; Jara-Ettinger et al., 2020), numbers (Feigenson et al., 2004; Hannagan et al., 2015; Spelke, 2017), space (Newcombe & Sluzenski, 2004; Bellmund et al., 2018), and social relations (Siegal & Varley, 2002; Scott & Baillargeon, 2017; Spelke, 2022). This "developmental start-up software" enables early learning (Spelke & Kinzler, 2007; Lake et al., 2017) and serves as the foundation for complex reasoning in variable environments later in life (Barsalou, 2020; Mitchell, 2021).

## 3. Benchmarking Core Knowledge in Multi-modal Large Language Models

We introduce **CoreCognition**, encompassing 12 core abilities and 1,503 questions with diverse input types and formats. An overview of the benchmark and its distribution is shown in Fig. 2, with 12 representative examples in Fig. 1. Sec. 3.1 outlines the cognitive taxonomy and theoretical framework guiding our benchmark. Sec. 3.2 details the curation process, while Sec. 3.3 describe model inference and evaluation.

### 3.1. Cognitive Framework

We take inspiration from Jean Piaget's theory (Piaget, 1950; Piaget & Inhelder, 1969; 1974), which identifies four stages in human developmental trajectory: Sensorimotor, Preoperational, Concrete Operational, and Formal Operational. In the Sensorimotor stage, infants develop core concepts such as object permanence (Spelke et al., 1992; Bremner et al., 2015) and perceptual constancy (Green, 2023) through sensory and physical interactions. The Preoperational stage serves as a transitional phase, characterized not by distinct new abilities but by the gradual solidification of symbolic representations (Fodor, 1975). These cognitive advancements culminate in the Concrete Operational stage, where children acquire abilities for systematic reasoning about numbers, motion, and agents, including perspective-taking,

conservation, intuitive physics, and hierarchical relations (Piaget & Inhelder, 1974; Moll & Meltzoff, 2011; Piloto et al., 2022; Murphy & Lassaline, 2013). The Formal Operational stage extends these abilities to abstract reasoning and complex tasks, such as understanding intentionality and mechanical reasoning (Kilner, 2011; Allen et al., 2020). See Appendix A.1 for empirical support of framework and Appendix A.2 for a detailed description of core abilities.

## 3.2. Dataset Curation

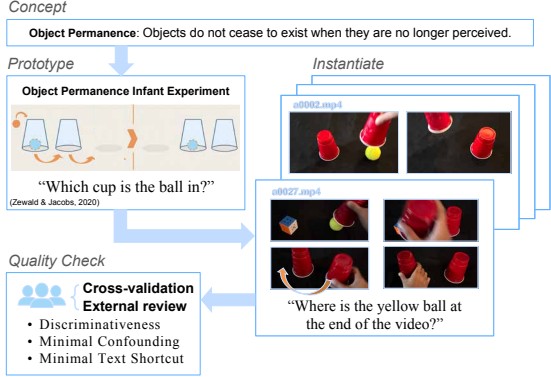

*Figure 3.* Overview of the benchmark curation process

Building upon the above cognitive framework, we operationalize theoretical constructs into explicit examples designed to probe specific core abilities in MLLMs. To ensure conceptual integrity and interdisciplinary rigor, we establish criteria that define successful instances: 1. *Discriminativeness*: Instances should be structured such that models lacking the targeted core knowledge necessarily select the incorrect answers, thereby ensuring the discriminative power. 2. *Minimal Confounding*: Questions should minimize reliance on confounding capabilities, such as object recognition, and must avoid conceptual overlap with other core knowledge included in the benchmark. 3. *Minimal Text Shortcut*: Instances should be crafted so that answers cannot be derived through textual shortcuts alone but require genuine multimodal comprehension.

A total of 12 annotators, each with a college-level education in cognitive science, computer science, or statistics, collaborate on the curation of **CoreCognition**.

**Prototyping.** We operationalize 12 theoretical concepts into a series of prototype scenarios that abstractly exemplify situations suited to evaluating specific core abilities within MLLMs. For example, to evaluate object permanence, we drew inspiration from classic infant experiments in developmental psychology (Zewald & Jacobs, 2020). The prototype scenario involves a ball being hidden under one of several cups, followed by occlusion and spatial manipulation. For each cognitive ability, we developed 5-10 prototype sce-

narios. These prototypes serve as templates from which concrete data instances can be generated.

**Instantiation of Prototypes.** To instantiate a prototype, we generate vision modalities (images/videos) that abstractly align with the underlying concept. These media assets are collected from a variety of sources, including internet, public datasets, synthetic content by generative models, simulated environments, and original recordings captured by cameras. Each asset is then paired with a carefully designed question that probes the specific core ability, along with a pre-defined options and the ground-truth answer, forming the multiple-choice questions (MCQs). Please refer to Appendix B for a detailed discussion on types and formats of curated questions and Appendix C for justification over the difficulty of the questions in **CoreCognition**.

**Quality Control.** Following the established criteria, each question-answer (QA) pair undergoes two rounds of independent cross-validation by annotators from separate groups. Any data point that fails to meet the standard is discarded. To further assess the reliability and clarity of the QA items, we conducted an additional round of validation by collecting responses from 20 human annotators via Amazon Mechanical Turk. We further recheck QAs that lead to consistent mistakes by humans.

## 3.3. Inference and Evaluation Strategy

Evaluating MLLMs on a large scale with QA formats poses several challenges: 1. MLLMs, ranging from 1B to 110B parameters, demand substantial computation resources and inference time with different environment dependencies for each of the 230 models, complicating efficient and robust evaluation under limited computational resources. 2. The free-form outputs from MLLMs can be highly variable, potentially leading to conceptual errors in performance assessment if an inappropriate evaluation method is used.

**Inference.** To address these challenges, we built a scalable evaluation infrastructure supporting parallel execution and compartmentalized environments, enabling reliable inference across over 200 MLLMs. We strictly follow the setup and source code from the official codebases provided by model developers to ensure fidelity. Further details are provided in Appendix D.1.

**Evaluation.** For each $k$-choice question, we cyclically rotate the answer options $k$ times, generating $k$ versions with different option orders. Rather than requiring consistent selection of the correct answer across all rotations to assign a correct score, we instead calculate the proportion of correct responses over the augmented set. This averaging approach avoids the exponentially diminishing chance-level accuracy that arises when enforcing consistency on questions with many options. We refer to Appendix D.2.3 for more details.

| Model | Sensorimotor | | | | | Concrete Operation | | | | Formal Operation | | | Mean |
|---|---|---|---|---|---|---|---|---|---|---|---|---|---|
| | Boundary | Continuity | Permanence | Spatiality | Perceptual Constancy | Intuitive Physics | Perspective Taking | Conservation | Hierarchical Relation | Intentionality Understanding | Mechanical Reasoning | Tool Using | |
| Human | 85.71 | 78.89 | 88.10 | 75.57 | 90.70 | 91.52 | 91.99 | 88.89 | 71.88 | 81.98 | 87.72 | 91.87 | 86.98 |
| *Proprietary Models* | | | | | | | | | | | | | |
| GPT-o1(Jaech et al., 2024) | 78.84 | 63.95 | 57.03 | 61.22 | **87.79** | **75.45** | **55.21** | 78.38 | **75.49** | **87.54** | 85.13 | 98.16 | **74.91** |
| GPT-4o(Hurst et al., 2024) | **87.17** | 63.37 | 52.86 | 67.95 | 81.20 | 58.79 | 43.06 | 69.97 | 57.52 | 85.20 | 77.51 | 96.91 | 69.25 |
| Qwen-VL-Max(Bai et al., 2023b) | 82.40 | 65.12 | 41.99 | 61.56 | 84.33 | 57.88 | 27.89 | 78.98 | 69.66 | 81.00 | **86.20** | 97.96 | 67.91 |
| Gemini-1.5-Pro(Team et al., 2024) | 82.58 | 69.19 | 59.56 | 67.95 | 73.26 | 56.21 | 30.96 | 64.26 | 73.62 | 81.00 | 80.56 | 98.09 | 67.80 |
| Gemini-1.5-Flash(Team et al., 2024) | 79.12 | **71.22** | 47.55 | 68.11 | 77.62 | 57.42 | 30.63 | 51.95 | 73.95 | 78.74 | 72.13 | 95.77 | 65.50 |
| GPT-4-Turbo(Achiam et al., 2023) | 77.62 | 61.34 | 48.94 | 54.25 | 82.17 | 63.48 | 43.11 | 50.15 | 51.62 | 80.92 | 75.81 | 98.75 | 65.23 |
| Grok-2-Vision(xAI, 2024) | 80.52 | 69.19 | 36.27 | 67.63 | 73.26 | 53.64 | 37.34 | 50.15 | 71.12 | 79.67 | 58.15 | 99.21 | 63.69 |
| Claude-3.5-Sonnet(Anthropic, 2024) | 78.28 | 56.40 | 32.43 | 64.74 | 76.74 | 58.18 | 34.17 | 46.55 | 64.56 | 72.98 | 69.00 | 97.96 | 61.92 |
| GPT-4o-mini(Hurst et al., 2024) | 74.25 | 70.06 | 38.89 | 53.93 | 64.73 | 59.39 | 41.01 | 42.04 | 58.66 | 76.01 | 69.35 | 96.12 | 60.89 |
| *Open Source Models* | | | | | | | | | | | | | |
| Qwen2.5-VL-72B(Bai et al., 2025a) | 79.59 | 64.53 | 43.38 | 62.82 | 85.27 | 59.09 | 28.91 | 81.68 | 68.20 | 80.45 | 85.66 | 97.96 | 68.29 |
| InternVL3-78B-Instruct(Zhu et al., 2025) | 80.62 | 58.86 | 64.37 | 58.90 | 79.73 | 52.12 | 70.79 | 73.64 | 40.20 | 34.03 | 54.99 | 91.18 | 64.60 |
| Ovis1.6-Gemma2-9B(Lu et al., 2024b) | 85.39 | 60.17 | 47.55 | 51.68 | 69.19 | 55.76 | 26.91 | 54.35 | 71.60 | 77.49 | 65.77 | 87.70 | 60.92 |
| mPLUG-Owl3(Ye et al., 2024) | 76.50 | 59.01 | 39.95 | 48.96 | 62.21 | 44.09 | 33.80 | 71.47 | 70.63 | 77.10 | 60.22 | 92.89 | 59.92 |
| Gemma-3-27B(Team et al., 2025a) | 78.18 | 50.00 | 35.05 | 49.86 | 60.47 | 51.52 | 35.01 | 53.75 | 69.74 | 77.55 | 70.53 | 97.52 | 59.38 |
| VILA1.5-40B(Lin et al., 2024) | 75.37 | 58.14 | 40.60 | 59.70 | 69.38 | 54.55 | 26.96 | 30.63 | 60.60 | 78.27 | 64.34 | 95.92 | 58.31 |
| DeepSeek-VL2(Wu et al., 2024b) | 76.69 | 65.41 | 35.87 | 55.21 | 69.09 | 57.42 | 31.98 | 32.43 | 64.72 | 80.14 | 48.84 | 92.52 | 58.17 |
| Pixtral-12B(Agrawal et al., 2024) | 72.28 | 55.52 | 44.28 | 49.84 | 66.86 | 51.97 | 25.19 | 58.26 | 59.06 | 70.87 | 58.42 | 96.91 | 57.78 |
| LLaVA-NeXT-72B(Liu et al., 2024a) | 79.21 | 65.12 | 61.03 | 47.84 | 68.60 | 52.27 | 30.12 | 38.44 | 60.11 | 77.65 | 70.07 | 46.21 | 56.25 |
| MMAlaya2(Ltd., 2024) | 77.81 | 66.86 | 48.28 | 42.47 | 72.67 | 59.39 | 28.12 | 28.83 | 65.94 | 81.07 | 64.25 | 46.86 | 55.19 |
| LLaVA-Onevision-Qwen2-72B-ov-hf(Li et al., 2024b) | 81.84 | 57.85 | 40.36 | 39.18 | 64.63 | 51.82 | 29.05 | 48.35 | 63.27 | 78.04 | 66.31 | 54.69 | 54.46 |
| Phi-4-multimodal-instruct(Abdin et al., 2024) | 78.57 | 52.59 | 31.60 | 36.02 | 63.57 | 53.64 | 25.75 | 51.95 | 60.42 | 67.29 | 65.77 | 75.46 | 53.55 |
| Idefics3-8B-Llama3(Laurençon et al., 2024) | 74.06 | 58.43 | 27.70 | 35.50 | 68.60 | 65.30 | 29.42 | 45.05 | 47.65 | 65.26 | 59.05 | 52.06 | 51.18 |
| Emu2-Chat(Sun et al., 2024) | 64.61 | 54.94 | 43.79 | 39.58 | 57.27 | 45.76 | 33.24 | 24.92 | 52.99 | 51.64 | 44.71 | 43.66 | 45.67 |

*Table 2.* Performance of selected MLLMs (classified into proprietary, open-source and reasoning models) on the **CoreCognition** dataset. Best results are shown in bold; second-best are underlined.

To assess free-form responses, we employ a two-stage scoring process. Multiple-choice questions (MCQs) inherently contain predefined answers and choices. Thus, each MLLM response is first mapped to one of the options or marked as *FAIL* if unaligned. Mapping is achieved via a hybrid method combining template matching with LLM-as-a-Judge. A pre-defined set of templates is utilized to directly match the model's output with one of the options. When template matching fails, an LLM will be used to determine the most suitable corresponding option. Models with a high *FAIL* rate are re-evaluated for validity and excluded from further analysis to avoid bias if they consistently produce nonsensical responses. In the second stage, the mapped choice is compared against the ground-truth answer, with *FAIL*s counted as incorrect. See Appendix D.2.1 for details, where we show consistent results with 4 alternatives.

# 4. Experiments

To comprehensively assess core knowledge in MLLMs, we meticulously selected and evaluated a diverse array of models across various architectures and scales. The evaluated set prominently includes commercial models such as the OpenAI and Claude series, high-performing open-source models like InternVL (Zhu et al., 2025) and the Qwen series (Bai et al., 2025b), as well as recently introduced models from the DeepSeek (Lu et al., 2024a; Wu et al., 2024a) series, which have garnered considerable attention. The evaluated open source models range in size from 1 billion to 110 billion, with different design choices such as dense (Vaswani et al., 2017; Devlin et al., 2019; Brown et al., 2020) and Mixture-of-Experts (Shazeer et al., 2017; Fedus et al., 2022), different vision encoders (e.g. Clip (Radford et al., 2021), SigLip (Zhai et al., 2023)). As discussed in Sec. 3.3 and Appendix D.2.1, we additionally filtered out models that consistently produce invalid outputs. Ultimately, we

have 230 models for subsequent analysis, among which 25 are proprietary, and 205 are open-source.

## 4.1. Core knowledge Deficits

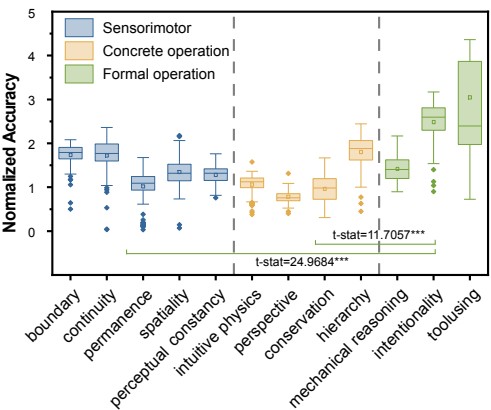

*Figure 4.* Accuracy by concept normalized by chance level. Evidence of core knowledge of deficits, with statistical significance.

As shown in Fig. 4, models exhibit a pronounced "core knowledge deficit": they perform significantly better on higher-level abilities (right side of the figure), comparable or even surpassing humans, but struggle with lower-level abilities (left side), associated with early developmental stages. This disparity is statistically significant and contrasts sharply with human performance, which remains consistently high across all stages. It's noteworthy that lower-level abilities are operational approximations of basic cognitive systems and are thus more directly aligned with the notion of "core knowledge", while higher-level abilities are more abstract or compositional cognitive tasks. The observed upward trend in performance does not imply that only lower-level abilities reflect core knowledge. Rather, it suggests that while mod-

els may perform better on higher-level tasks—potentially by pattern matching or spurious correlation—they often struggle with the more fundamental reasoning required for lower-level tasks. This gap implies the failure to demonstrate a robust understanding of the foundational abilities that higher-level tasks presuppose. Details on Fig. 4, accuracy normalization, and pairwise t-tests are provided in Appendix E. We further validate whether the observation holds under a variety of different conditions in Appendix H.

Tab. 2 compares the performance of 30 state-of-the-art MLLMs with human performance. All MLLMs substantially underperform relative to humans on lower-level stages (i.e. Sensorimotor and Concrete Operation excluding Hierarchy): the best models, GPT-o1, GPT-4o and Qwen2.5-VL, achieve average scores of 74.91%, 69.25%, and 68.29% trailing human performance by 15.91%, 21.57%, and 22.53%, respectively. Notably, proprietary models do not consistently outperform open-source counterparts; for instance, GPT-4o outperforms Qwen2.5-VL-72B by only 1% on average and Gemini and Claude series underperform Qwen2.5-VL-72B, QVQ-72B-Preview and InternVL3-78B by 2-3% on average. This indicates that both proprietary and open-source MLLMs share the core knowledge deficit, underscoring a fundamental limitation across all models. Particularly, models perform markedly worse than humans on Perspective Taking, likely reflecting their limited capacity for mental simulation, critical for understanding alternative viewpoints (Barnes-Holmes et al., 2004; Moll & Meltzoff, 2011; Barlassina & Gordon, 2017). This highlights broader concerns regarding the absence of robust model-based reasoning in contemporary language models (Lake et al., 2017; Mitchell & Krakauer, 2023).

> **Key Finding 1 (Core Knowledge Deficits):** MLLMs excel at higher-level abilities associated with later developmental stages but consistently struggle with lower-level abilities that typically emerge earlier in human cognition.

### 4.2. Dependencies Between Core Abilities

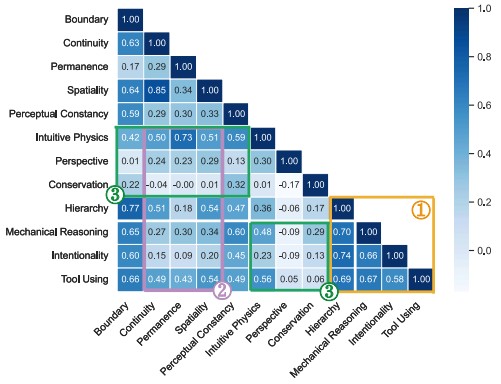

*Figure 5.* Pearson Correlations Between Core Abilities.

Examining the interdependencies among core abilities provides a principled understanding of whether models develop coherent, hierarchically structured competencies akin to those seen in humans. To quantify the degree of covariation consistent with developmental hierarchies, we compute Pearson correlations between performances across all 12 abilities. The results reveal a distinct divergence: many correlations are modest ($\rho < 0.4$), while some clusters exhibit strong alignment ($\rho > 0.65$). As illustrated in Fig. 5, we observe ① robust correlations among several high-level abilities, reflecting the anticipated interdependence among tasks within the same developmental stage—a pattern notably absent in earlier stages of model performance. Notably, the Hierarchy ability clusters more closely with Formal Operational abilities, consistent with its strong overall performance and suggesting that models may treat it as an advanced reasoning task. In contrast, ② three Sensorimotor abilities (Permanence, Spatiality, and Continuity) exhibit weak correlations with most higher-stage abilities, implying that these foundational competencies do not provide the developmental scaffolding to more advanced stages, which are typically observed in humans. ③ Further evidence can be observed from three Concrete Operational abilities (Perspective, Conservation, and Intuitive Physics), which also show weak cross-stage correlations. Collectively, these results indicate that current models lack structured representational dependencies, raising concerns about the grounding and internal coherence of their acquired abilities(Spelke et al., 1992). We further validate whether the observation in Fig. 5 holds under a variety of conditions in Appendix I.

> **Key Finding 2 (Misaligned Dependency):** Core abilities exhibit weak cross-stage correlations, indicating an absence of developmental scaffolding.

### 4.3. Core Abilities are Predictive of Higher-level Abilities

To support the argument that core knowledge is essential for higher-level reasoning and perceptual abilities, we show in Fig. 6 that strong performance on core abilities reliably predicts higher performance on most high-level abilities and benchmarks, such as SEEDBench2 (Li et al., 2024a). Concretely, we analyze the correlation between the performance of 12 core cognitive concepts across three stages and the performance of the same models on 26 public benchmarks and 9 higher-level abilities, as defined by SEED-Bench 1 (Li et al., 2023a) and 2(Li et al., 2024a). Our findings reveal that, except for perspective and Intuitive Physics, core abilities strongly predict performance on public benchmarks (except ChartQA) and performance of higher-level abilities in SEED-Bench. We hypothesize that the exceptions of ChartQA arise because textual understanding is largely orthogonal to the core abilities examined here. Perspective and Intuitive Physics tasks demand structured internal

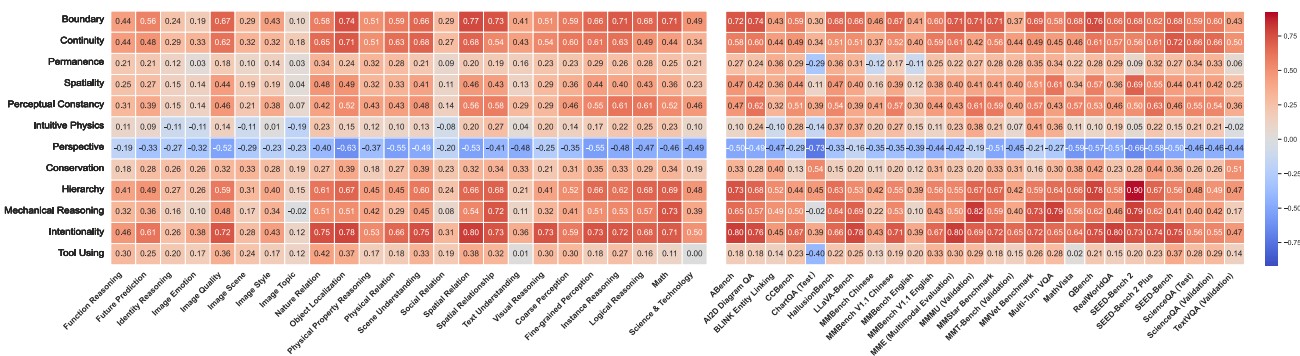

Figure 6. Correlation between core abilities and "high-level" abilities (left) and existing MLLM benchmarks (right).

representations and counterfactual reasoning—abilities that underpin advanced reasoning in humans. The observed lack of correlation indicates that core knowledge deficits, as evidenced by dependencies between different abilities, are also reflected in benchmark evaluations. We further validate whether the observation in Fig. 6 holds under a different conditions in Appendix J.

> **Key Finding 3 (Predictability):** Performance on core knowledge is predictive of higher-level abilities.

## 4.4. Scaling Effect on Core Knowledge?

Not for low-level abilities! The advancement of LLMs has been driven by the empirical scaling law—predictable power—predictable power-law improvements in performance with increased compute, parameters, and training data (Kaplan et al., 2020; Zhai et al., 2022; Henighan et al., 2020; Hoffmann et al., 2022)—and emergence, the abrupt appearance of qualitatively new abilities as model scale increases (Wei et al., 2022a; Aghajanyan et al., 2023; Bubeck et al., 2023; Berti et al., 2025). This raises a fundamental question: *Does performance on core knowledge also emerge and scale as models increase in size?* We evaluate the extent to which scaling applies to low-level core abilities rooted in core knowledge. By fitting a linear regression to the performance of 230 models of varying sizes on these abilities, we estimate the scaling effect as the slope of the regression line. As shown in Fig. 7, our results reveal a clear dissociation between low- and high-level abilities regarding scaling effects. For seven out of nine low-level abilities—excluding hierarchical relation and perceptual constancy—in the Sensorimotor and Concrete Operational Stages, model performance shows significantly less improvement with increasing size, compared to the higher-level Formal Operational Stage. Notably, perspective-taking ability even declines with scale, likely due to a persistent egocentric bias that intensifies as models grow larger. These findings indicate that scaling primarily benefits high-level reasoning, while its impact on low-level cognitive abilities is limited or even negative. This suggests that simply increasing model size is insufficient

for developing core knowledge in MLLMs. We further discuss the generalization of this conclusion under different conditions in Appendix J.

> **Key Finding 4 (Not Scaling):** MLLMs exhibit limited or no scalability on low-level abilities compared to high-level abilities

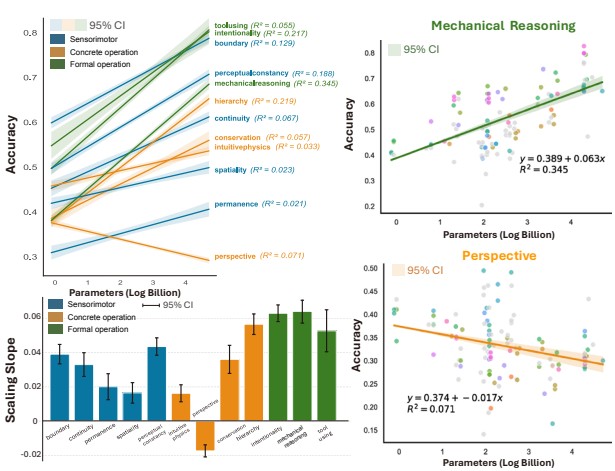

Figure 7. Scaling Effect on core knowledge with respect to model size. Scaling laws do not apply uniformly across all concepts. **Top Left.** Fitted curves for each concept across 219 models and 11 prompt cases. **Bottom Left.** Comparison of core abilities' scalability (slopes of fitted curves). **Right.** Scaling curve for Mechanical Reasoning/Perspective-taking. Dots of the same color represent models from the same series.

## 4.5. Does Reasoning Help?

Reasoning and test-time scaling are widely adopted in MLLMs and have demonstrated strong performance on complex benchmarks such as MathVista (Lu et al., 2023), CLEVR (Johnson et al., 2017), and Geometry3K (Lu et al., 2021). One might hypothesize that these approaches enable more effective knowledge structuring, thereby improving performance on core knowledge tasks such as those in our benchmark. For example, in the "near-large-far-small" perceptual constancy scenario shown in Fig. 9, models relying

on system-1 thinking may naively predict that the bridge narrows from direct perception, whereas models employing system-2 reasoning could override this intuitive illusion by more thinking and correctly infer that the apparent narrowing is merely a result of perspective.

To examine whether reasoning and test-time scaling enhance performance on core cognitive abilities, we evaluate both reasoning-augmented models and their corresponding instruction-tuned counterparts on **CoreCognition** questions including Kimi-VL-A3B-Thinking, Kimi-VL-A3B-Instruct (Team et al., 2025b), QVQ-72B-Preview (Team, 2024b), R1-Onevision-7B (Yang et al., 2025b), Llama-3.2V-11B-cot (Xu et al., 2024), Llama-3.2-11B-Vision(Meta, 2025), as well as full series of InternVL3 (Zhu et al., 2025) and VLAA-Thinker-Qwen2/2.5VL (Chen et al., 2025) to Qwen2/2.5-VL (Wang et al., 2024a; Bai et al., 2025b).

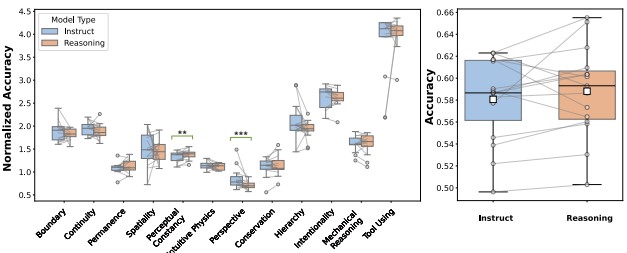

*Figure 8.* **Left.** By concept comparison between reasoning models and their non-reasoning counterparts. **Right.** Comparison of the overall performance

As per Fig. 8, reasoning abilities and test-time scaling do not confer a clear advantage over instruction-tuned models. Overall, reasoning models show a modest, non-significant average improvement. The only two exceptions fail to exhibit a consistent trend (perceptual constancy, where reasoning models perform better ($P = 0.0669$), and perspective taking, where they perform worse ($P = 0.0037$)). The absence of improvement in lower-level abilities underscores that current architectures struggle with grounded reasoning. Similarly, the negligible gains observed on higher-level tasks such as Intentionality Understanding and Tool Use suggest limited advantages from reasoning at near-ceiling performance. In contrast, reasoning models do not improve the relatively low scores in Mechanical Reasoning, likely because this domain requires model-based reasoning that cannot be addressed by chain-of-thought prompting or test-time scaling (Hegarty, 2004; Mitchell, 2021). Notably, reasoning-augmented models exhibit a narrower performance distribution, indicating that explicit reasoning may stabilize outputs without resolving deeper representational deficits.

## 5. *Concept Hacking*: A Controlled Experiment

*Do MLLMs genuinely possess core knowledge?* A fundamental challenge in evaluating the abilities of MLLMs is

their propensity to exploit spurious features, where apparent task proficiency may stem from shortcut learning rather than genuine understanding (Alvi et al., 2018; Bahng et al., 2020; Cadene et al., 2020; Clark et al., 2019; Dagaev et al., 2021).

### 5.1. Methodology

We introduce *concept hacking*, which systematically manipulates task-relevant features while preserving task-irrelevant conditions to completely invert ground truth labels. As exemplified in Fig. 9, 45 samples from CoreCognition are paired with a manipulated version containing identical questions but opposite correct answers. Given a pair of tasks, it yields four possible MLLM response types (Tab. 3): The case where models answer controlled tasks incor-

*Table 3.* Four types of outcomes from MLLM.

| Control | Manipulation | Interpretation |
|---------|--------------|----------------|
| ✓ | ✓ | core knowledge |
| ✓ | ✗ | shortcut |
| ✗ | ✓ | core deficits |
| ✗ | ✗ | |

rectly but manipulation tasks correctly reflects coincidental accuracy—models lacking core knowledge produce wrong answers on controlled tasks, but inverted ground truth in manipulation tasks makes incorrect reasoning appear correct, i.e. "being right for the wrong reason". See a detailed explanation of concept hacking curation in Appendix K.

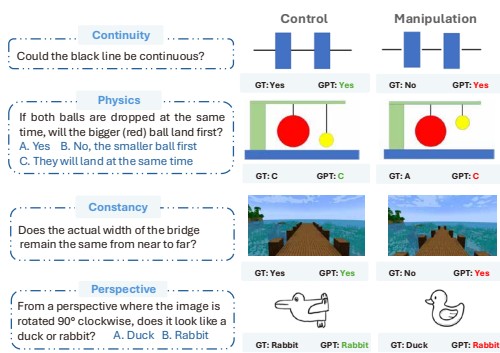

*Figure 9.* Examples of Concept Hacking.

### 5.2. Results: Core Deficits v.s. Shortcut Taking

The results reveal a clear separation between models exhibiting shortcut and those with core deficits (Fig. 10), with A substantial proportion of models clustered in the top left quadrant (high manipulation accuracy, below-chance control accuracy, consistent with findings in Sec. 4.1), and a significant portion of models appeared in the bottom right quadrant (high control accuracy, below-chance manipulation accuracy), reflecting a pronounced reliance on shortcuts with a high susceptibility to manipulation. Most models demonstrated above-chance performance on both tasks but still fell short of human-level core knowledge. Unlike humans, almost none of these models achieved roughly equal accuracy

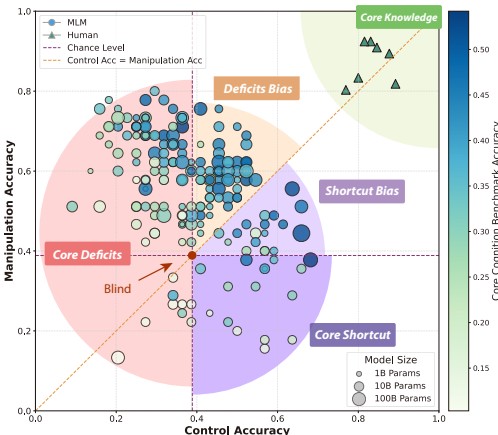

*Figure 10.* Control vs. Manipulation accuracy. Circle: model performance; Size of circle: parameter size of model; Color of circle: averaged accuracy on **CoreCognition**; Green triangle: human performance; Red dot: chance level point on both Control and Manipulation, termed as "blind". As size increases, core deficits and shortcut reliance intensify, rather than progressing toward the human-like core knowledge region.

on both tasks—a hallmark of robust core knowledge and immunity to *concept hacking*. This pattern suggests that, while some models are not entirely dominated by shortcut-taking or lack of core knowledge, these factors still substantially affect their predictions. Interestingly, susceptibility to *concept hacking* does not correlate straightforwardly with model size or performance on **CoreCognition**. While many shortcut-reliant models were smaller and weaker, the bottom right quadrant also include some of the largest and best-performing models, such as GPT-4o. Similarly, models with core deficits in the top left quadrant varied in size and performance. In line with previous findings on the non-scaling of low-level abilities, increasing model size does not inherently improve core knowledge, but rather enhances shortcut-taking or the persistence of core deficits.

> **Key Finding 5 (Deficits v.s. Shortcut Taking):** Models increasing in size exhibit deficits and shortcut-taking behaviors rather than progressing toward conceptual understanding of core knowledge.

## 6. Discussion

Our findings (1–3) support the hypothesis that MLLMs lack core knowledge that grounds the high level perception and reasoning abilities, and that such core abilities cannot be acquired through scaling alone (4–5). This offers an explanation for the longstanding Moravec's paradox that tasks intuitive and effortless for humans often prove to be the most challenging for machines (Moravec, 1988). It also offers a plausible account for the lack of robustness in current MLLMs (Shiffrin & Mitchell, 2023; Zhang et al.,

2024b), and resonates with ongoing critiques that foundation models fail to develop genuine conceptual understanding, instead reinforcing shortcut-based strategies as they scale (Bender et al., 2021; Mitchell & Krakauer, 2023).

Our findings that current training paradigms fail to instill core cognitive abilities underscore the need for pretraining strategies that explicitly target these foundational capacities. More specifically, if core knowledge cannot be directly acquired through scaling, it may be beneficial to first teach or distill core knowledge into MLLMs prior to large-scale pretraining, thereby enabling more data-efficient generalization akin to human learning. From both an evaluation and design perspective, our benchmark reveals distinct failure modes—including deficits in permanence, spatiality, boundary, and continuity. These limitations further hinder abilities such as visual perspective-taking and contribute to an over-reliance on shortcuts, which is a fundamental cause of poor out-of-distribution generalization.

One possible objection is that human-like core knowledge is not essential for artificial general intelligence (AGI), even when AGI is defined in relation to human-level intelligence. Intelligence, after all, may be multiply realizable—achievable through architectures and developmental paths distinct from those of humans (Bechtel & Mundale, 1999). However, core knowledge may embody fundamental learning principles that recur across intelligent agents, including non-human animals (Santos, 2004; Lake et al., 2017). If such a theory holds, non-human pathways to AGI, e.g. scaling, will also lead to the emergence of these core abilities. In this light, a benchmark on core knowledge, such as ours, could offer a useful lens for evaluating the progress toward AGI, regardless of the path taken. In light of ongoing uncertainty about the path towards AGI, the human developmental trajectory offers a valuable, empirically grounded reference point. Despite great advancements, MLLMs consistently struggle with hallucinations, poor generalization, and a lack of robustness, suggesting that key cognitive ingredients may still be lacking. By aligning evaluation with structures known to support robust reasoning and perception in humans, our framework helps expose and address these critical gaps in emerging AI systems.

## 7. Conclusion

We introduce **CoreCognition** benchmark paired with a novel *Concept Hacking* method to rigorously and controllably evaluate the conceptual understanding of core knowledge in MLLMs. We uncover four key findings that collectively indicate a consistent lack of core knowledge in current MLLMs—fundamental understanding of basic world concepts such as objects, actions, numbers, space, and social relations—that humans acquire from infancy. We also discuss potential limitations in Appendix M.

## Impact Statement

We aim to advance the development of MLLMs through a cognitively grounded evaluation framework, **CoreCognition**. Our findings reveal that MLLMs lack conceptual understanding of core knowledge, cautioning against overinterpreting their success on complex tasks. The curation method employed in **CoreCognition** offers a scalable methodology that may inform future benchmarking of foundation models in large scale. Our findings also shed light on the design of more robust and interpretable models with grounded reasoning and perception capabilities. The adversary idea in *Concept Hacking* may pose misuse risks if adapted for military purposes.

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

# A. Cognitive Framework

## A.1. Core Knowledge in Human

Past research has shown that humans exhibit a series of rudimentary yet robust abilities in domains such as object, number, space, action, and social cognition at a very young age. Such abilities, often known as "core" cognition, ground the set of diverse and complex abilities of human intelligence that develop later (Spelke et al., 1992; 1994; 1995; Spelke & Kinzler, 2007; Baillargeon & Carey, 2012; Mitchell, 2020; 2021). From infancy to early adulthood, human cognition develops along a structured trajectory, with interdependent relations between early, simple abilities and late, complex abilities. For instance, the ability to imagine the perspectives of others typically develops between the ages of 3 and 6 (Piaget & Inhelder, 1969), while the capacity to fully comprehend others' intentions matures around age 12 (Wimmer & Perner, 1983; Wellman et al., 2001; Liu et al., 2008). At the same time, the ability to understand other people's intentions largely depends on the ability to understand other people's perspectives (Iacoboni, 2009; De Waal & Preston, 2017; Liu et al., 2017; Caviola et al., 2021; Ninomiya et al., 2020). An influential account of human learning has suggested that cognitive development is fundamentally driven by the increase of computational/representational power of the system, which allows for more complex mental operations to be performed on external data (Fodor, 1975; Pylyshyn, 1980; Halford et al., 1998; Fodor, 2008). However, while high-level abilities might emerge directly due to enhanced operational resources, these operations are critically guided by the "core" cognition system that has enabled the system to possess a rudimentary understanding of each cognitive domain. This early-stage grounding not only empowers humans to achieve a reliable performance at basic yet widely-applicable tasks starting from very young ages but is also precisely what supports high-level abilities to robustly direct task-relevant behaviors despite the nuanced signals that exist in the environment (Mitchell, 2021).

The sensorimotor stage is the first stage of cognitive development proposed by Jean Piaget (Piaget, 1952; Piaget & Inhelder, 1974). Spanning from birth to approximately 2 years of age, this stage is characterized by infants' understanding of the world through their sensory experiences and motor actions. Several prominent features of human intelligence developed during this period. First, infants develop object permanence, that they realize objects and people continue to exist even when not in direct sight, or being heard or touched (Baillargeon et al., 1985). They start to understand that there is a sense of continuity for the ways that objects exist, and the inductive bias of continuity is essential, e.g., for recognizing objects when occluded or for continuously tracking objects (Spelke et al., 1995; Le Poidevin, 2000). Infants also develop the sense of boundary during this stage, namely, the ability to recognize where one object ends and another begins (Kestenbaum et al., 1987; Jackendoff, 1991). Lastly, infants develop spatial and perceptual constancy by the end of the sensorimotor stage. Spatiality refers to the ability to perceive the position and distance of objects relative to oneself and each other, and recognize the spatial invariance between them when presented by various sensory experiences (Hermer & Spelke, 1996; Bell & Adams, 1999).

The preoperational and concrete operational stages are the second and third stages of Piaget's cognitive development. Typically spanning over 2 to 7 years of age, the preoperational stage is the transitional stage to the concrete operational stage, which children enter around 7 years of age. During this period, children begin to develop internalized mental actions supported by organized structures that can be manipulated and reversed in systematic ways, known as mental operations (Janet, 1905; Kirkpatrick, 1908; Piaget, 1950; Piaget & Inhelder, 2014; Miller, 2016). Through mental operations, children are then able to rigidly perform tasks that are previously unreachable, such as thinking from other people's perspectives, understanding hierarchical relations of objects, and reasoning about physical events in the world. These tasks require not only rudimentary understandings of physical concepts, which gradually became in place during the preoperational stage, but also relational and transformational reasoning that can only be done through mental operations (Piaget & Inhelder, 1974; Church & Goldin-Meadow, 1986; Houdé, 1997). Since the preoperational stage is mostly meaningful as the transitional period preceding the concrete operational stage, we do not have evaluation dimensions specifically targeting the stage. However, tasks targeting the concrete operational stage could assess the existence of knowledge associated with the preoperational stage, such as the law of conservation (Piaget, 1952; Halford, 2011; Houdé, 1997).

The formal operational stage is the fourth and final stage in Piaget's theory of cognitive development, typically emerging around 11 or 12 years of age and continuing into adulthood (Inhelder & Piaget, 1958). Starting in this stage, one is able to systematically and flexibly apply mental operations to not only concrete, physical domains but also abstract, formal domains (Kuhn & Angelev, 1976; Shayer, 1979; Huitt & Hummel, 2003). In particular, this stage is characterized by the development of complex thinking and reasoning abilities, such as abstraction, pattern recognition, the use of logic, and hypothetical and counterfactual reasoning (Piaget, 1950; Inhelder & Piaget, 1958). These cognitive advances pave the way for more sophisticated abilities to interact with the physical world, marked by mechanical reasoning and tool use (O'Brien & Shapiro,

1968). Together, there is an advance in social cognition, characterized by a deeper understanding of intentions, actions, and the reasoning behind them (Meltzoff, 1999).

## A.2. Definition of the 12 Core abilities in CoreCognition

**Boundary**  Boundary refers to the cognitive understanding of where one object ends and another begins, an essential aspect of perceiving and understanding the physical world (Kestenbaum et al., 1987). Without understanding boundaries, it seems very hard to construct a concept of the object (Berkeley, 1709; Jackendoff, 1991).

**Spatiality**  Spatiality refers to the cognitive understanding of the topological properties of our physical world (Bell & Adams, 1999). In a classic A-not-B task, an object is hidden at location A (such as under a cup) and the child successfully finds it several times. Then, the object is visibly moved to a different location B (under a different cup), in full view of the child. Younger infants often make the error of searching for the object at the original location A, indicating a developmental stage where their understanding of object spatiality is not yet formed.

**Perceptual Constancy**  Perceptual constancy is the cognitive ability to perceive objects as being constant in their properties, such as size, shape, and color, despite changes in perspective, distance, or lighting (Rutherford & Brainard, 2002; Khang & Zaidi, 2004; Green, 2023). For instance, consider a red ball being thrown in a park. To an observer, the ball appears smaller as it moves farther away, yet the observer understands it remains the same size throughout its trajectory.

**Object Permanence**  Object permanence refers to the cognitive understanding that objects continue to exist even when they are no longer perceptually accessible (Baillargeon, 1986; Spelke et al., 1992). This capacity emerges early in infancy and marks a shift from sensorimotor interactions to rudimentary conceptual reasoning. A classic example is peek-a-boo: initially, infants may react with surprise or distress when a caregiver's face is covered, as if it has ceased to exist. As permanence develops, they begin to infer the face's continued presence—reflecting the emergence of internal object representations that persist beyond immediate perception.

**Continuity**  Continuity is the cognitive prior that objects persist as unified, cohesive entities across space and time (Spelke et al., 1995; Le Poidevin, 2000; Spelke et al., 1994; Yantis, 1995; Yi et al., 2008; Bertenthal et al., 2013). For example, when we see the front and rear of a train simultaneously extending from opposite ends of a tunnel, we infer that the train continues through the occluded space as a single, continuous object. This inference reflects our sensitivity to spatiotemporal continuity—not merely that the object exists, but that its parts remain connected along a coherent trajectory through space.

**Conservation**  Conservation refers to the ability to understand that certain properties of physical entities are conserved after an object undergoes physical transformation (Piaget & Inhelder, 1974). This is instantiated in their ability to tell that quantities of physical entities across different domains, such as number, length, solid quantity and liquid volume, will remain the same despite adjustments of their arrangement, positioning, shapes, and containers (Halford, 2011; Craig et al., 1973; Piaget & Inhelder, 1974; Houdé et al., 2011; Poirel et al., 2012; Marwaha et al., 2017; Viarouge et al., 2019). For example, when a child watches water being poured from a tall, narrow glass into a short, wide one, a grasp of liquid conservation would lead them to understand that the amount of water remains the same even though its appearance has changed.

**Perspective-taking**  Perspective-taking is the ability to view things from another's perspective. This ability has seminal importance both to the understanding of the physical world as well as to the competence in social interactions (Wimmer & Perner, 1983; Wellman, 1992; Liu et al., 2008; Barnes-Holmes et al., 2004). The Three Mountain Task first invented by Jean Piaget is widely used in developmental psychology laboratories as the gold standard for testing perspective-taking abilities in children (Piaget & Inhelder, 1969)

**Hierarchical Relation**  Hierarchical relation refers to the ability to organize objects or concepts into structured categories and subcategories, which are supported by the development of mental operations marked by class inclusion and transitivity (Shipley, 1979; Winer, 1980; Chapman & McBride, 1992). Class inclusion refers to the ability to recognize that some classes or groups of objects are subsets of a larger class. For example, a child in the concrete operational stage is able to understand that all roses are flowers, but not all flowers are roses (Borst et al., 2013; Politzer, 2016). This concept is essential for one's systematic and logical organization of conceptual knowledge. Transitivity refers to the ability to understand logical sequences and relationships between objects (Andrews & Halford, 1998; Wright & Smailes, 2015). For instance, if a child knows that Stick A is longer than Stick B, and Stick B is longer than Stick C, they can deduce that Stick A is longer than Stick C.

**Intuitive Physics**  Intuitive physics refers to the ability of humans to predict, interact with, and make assumptions about the

physical behavior of objects in their world (Michotte, 1963). As children grow, they transition from simplistic understandings, such as expecting unsupported objects to fall, to more complex theories, such as grasping the principles of inertia (Spelke et al., 1994; Kim & Spelke, 1999) and gravity (Vasta & Liben, 1996; Kim & Spelke, 1999; Li et al., 1999).

**Intentionality Understanding** Intention understanding involves recognizing and interpreting the actions of others (Searle, 1979; Rosenthal, 1991). This process is not just about observing a behavior but also about understanding the goal behind it (Baker et al., 2009; Gandhi et al., 2021). For example, seeing someone reaching for a cup is not just about recognizing the physical action but understanding the intention behind it (e.g., they want to drink).

**Mechanical Reasoning** Mechanical reasoning refers to the ability to understand and apply mechanical concepts and logical principles to solve problems (Allen et al., 2020). This cognitive concept first involves the ability to interpret and predict the behaviors of complex physical systems and understand how different mechanisms of the systems work. Second, mechanical reasoning requires the ability to apply logic rules, such as induction, abduction, syllogism (O'Brien & Shapiro, 1968; Cesana-Arlotti et al., 2018), and reasoning forms, such as hypotheticals and counterfactual (Byrne, 2016), to figure out how to manipulate these systems to achieve a desired outcome (Hegarty, 2004).

**Tool Using** Tool-using refers to the ability to utilize objects (as tools) in their environment as aids in achieving a specific goal, such as obtaining food or modifying the surroundings. A lot of cognitive components are involved in tool-using ability, such as affordances, referring to computing the action possibilities offered to the agent by the tool with reference to the agent's sensorimotor capabilities (Gibson, 1979). For example, a door handle affords pulling or pushing, as how the door should be operated by a human agent.

## B. Input Types and Formats

**CoreCognition** encompasses diverse input types and formats. We first introduce two types of questions i.e, true/false questions and multiple-choice questions (MCQs). The distribution statistics is in Fig. 11.

Evaluating core knowledge presents unique challenges, as different core abilities necessitate distinct forms of visual media. For instance, tool use tasks require only images, while conservation tasks require videos. To address this diversity, our benchmarks incorporate a variety of visual formats within a single QA setting, including single images, videos, and sets of multiple images.

However, not all Multi-modal Large Language Models (MLLMs) can process every input format due to implementation constraints. Considering the more than 200 models evaluated in this study, we classify the input formats into three main categories: single image, single video, and multiple images (i.e., multiple frames). The distribution of these formats across our benchmark is shown in Fig. 11.

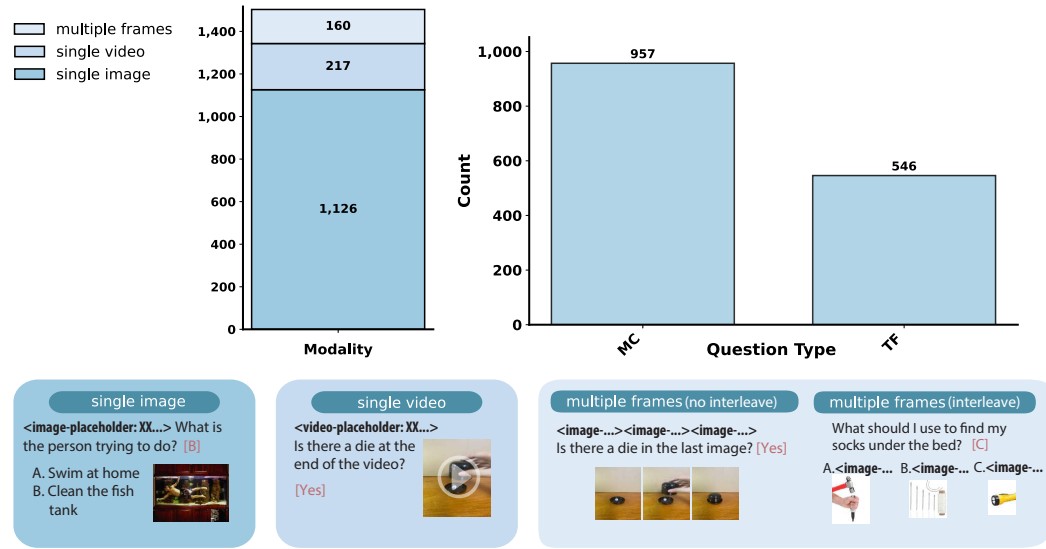

*Figure 11.* Distribution of question type and modality type in **CoreCognition** dataset.

## C. Justification for the Difficulty in CoreCognition

Our benchmark is deliberately designed to be more challenging than tasks typically used in classic developmental cognitive science experiments with young children, driven by two main considerations:

- **Breadth and systematic task coverage.** Each task in our benchmark is derived from a well-established developmental cognitive science prototype (e.g., Piaget's Three-Mountain perspective-taking paradigm). Rather than relying on the limited examples available in literature, we substantially expand each prototype into a comprehensive suite of systematically varied questions. This approach produces a rich and robust evaluation corpus, essential for reliable and nuanced measurement.

- **Appropriate evaluation for MLLMs.** Appropriate evaluation for large language models. The subjects of our evaluation are not infants or young children, but MLLMs adapted from SOTA LLM, with knowledge and reasoning abilities comparable to, or exceeding, those of well-educated adults, including PhD-level expertise in many domains (OpenAI, 2024). Consequently, our tasks are intentionally challenging to thoroughly probe the capabilities of these models, rather than simply replicating child-level diagnostic tasks.

## D. Inference and Evaluation

### D.1. Model Inference

We evaluate a total of 231 models, including commercial models and open-source models. Our tested models exhibit diversity in architecture and size, ranging from 1B to 110B parameter size. Inference is performed on clusters equipped with 8×NVIDIA A100 80 GB GPUs. In most cases, models between 1B and 13B in size can be inferred on a single GPU. Models ranging from 13B to 32B require two GPUs, those from 32B to 70B require four GPUs, and models larger than 70B require all eight GPUs to inference. Based on the input types they support, the 231 models are categorized into three groups: single-image, multi-image, and video models. Specifically, 85 models support only single-image input, 105 models support multi-image input, and 41 models support video input. We built a scalable evaluation infrastructure supporting parallel execution and compartmentalized environments, enabling reliable inference across over 200 MLLMs. We strictly follow the setup and source code from the official codebases provided by model developers to ensure fidelity. We further conduct sanity checks using widely adopted benchmarks to verify that the models can reproduce established results. To support smooth inference of over 200 models, we configure 43 compartmentalized environments, each compatible with one or multiple models. Efficient inference is performed by parallelizing models across multiple GPUs and devices. A dynamic scheduler is employed to minimize computational waste.

### D.2. Evaluation

D.2.1. MATCHING ANSWERS TO CHOICES

We explore four matching strategies and propose a hybrid approach that combines the strengths of both template- and LLM-matching. After removing pre-defined special tokens,

- **exact matching** matches the MLLM output to a choice only if they are identical, disregarding case differences.

- **"in" matching** matches the MLLM output to a choice if the output, when split by spaces or punctuation, contains exactly one choice.

- **template matching**: matches the entire MLLM output against predefined templates, such as "Answers: [choice]" or "[choice]. [sentences of explanation without references to another choice]".

- **LLM matching**: We employ a LLM-as-a-judge framework where the LLM is provided with the original question, the options, and the MLLM output and prompts it to determine which choice the output mostly supports.

Exact and "in" matching approaches exhibited relatively high fail rates, with abundant false positives and false negatives. These methods often struggle when model outputs are complex, such as reasoning models, or with explanation or chain-of-thought (CoT) responses. Template matching was able to accommodate a broader range of scenarios but necessitated iterative adaptation of templates to cover exceptional cases. Even after considerable refinement and despite achieving high

accuracy on matched data points, template matching still resulted in a non-negligible overall fail rate. In contrast, LLM matching demonstrates strong performance in identifying the intended choice within text-rich outputs, even in cases where the explanation underwent concession processes. However, LLM matching was occasionally susceptible to hallucination, particularly when short or simple answers were embedded within extensive contextual information.

To address these limitations and capitalize on the complementary strengths of different matching strategies, we propose a *Hybrid Matching* mechanism. Specifically, we prioritize a rule-based template matching approach to extract answers from MLLM responses. If template matching method failed, we turn to a model-based ensemble strategy using four advanced LLMs: Qwen2.5-72B-Instruct, Mixtral-8x7B-Instruct-v0.1, DeepSeek-R1-Distill-Llama-70B, and llama3.1-70B. The LLM-based result is accepted only when at least three of the four models produce consistent extractions; otherwise, the matching is deemed unsuccessful. By integrating the precision of regular-format matching with the flexibility of semantic-based matching, the hybrid method achieves more robust and reliable performance overall. We provide a comparison between *Hybrid Matching* and the other four alternatives in Tab. 4.

*Table 4.* Fail rates across different matching strategies.

| Method | Exact Matching | "In" Matching | Template Matching | LLaMA3.1-70B Matching | Hybrid Matching |
|---|---|---|---|---|---|
| **Fail Rate** | 53.0448% | 31.3501% | 8.2056% | 10.9640% | 6.4845% |

### D.2.2. FILTERING

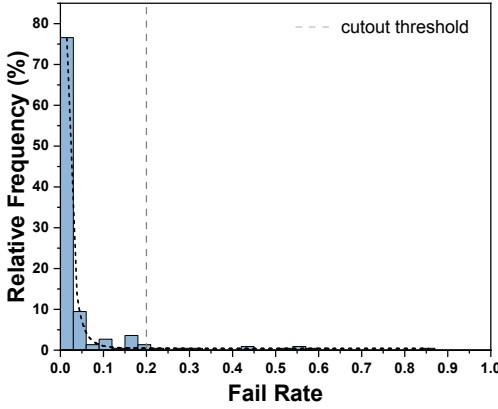

*Figure 12.* Distribution of fail rate in responses matching. We cut off at 20% filtering any models with fail rate over 0.2.

After applying *Hybrid Matching*, a number of models still exhibited high failure rates, as shown in Fig. 12. The distribution of failure rates across models revealed a long-tail pattern, with a small subset of models performing substantially worse than the majority. To distinguish between detrimental or systematic failures (e.g., outputs consisting entirely of illegal characters) and intrinsic model limitations (e.g., adequate input processing but inadequate responses), we manually examined all models with a matching fail rate of $\geq 17\%$. This thorough review allowed us to establish a clear threshold between these categories. Based on our analysis, we set a final exclusion criterion of $\geq 20\%$ fail rate, resulting in the removal of 12 models exhibiting detrimental failure modes. The remaining 219 models, which demonstrated reasonable performance, were retained for further analysis.

### D.2.3. SCORING

After applying *Hybrid Matching* and filtering out models with high failure rates, we evaluated each model by comparing its matched response to the ground-truth options. Responses marked as matching failures were classified as incorrect.

To reduce the risk of models favoring certain answer positions—a phenomenon known as option-position bias—we use the circular evaluation strategy (Liu et al., 2023a). In this approach, each multiple-choice question with $k$ possible answers is presented $k$ times, with the order of the answer options rotated each time.

Instead of requiring the model to pick the correct answer every time, regardless of the order, we calculate the proportion of times it selects the correct answer across all rotations. This method provides a fairer estimate of the model's true ability and avoids unfairly lowering scores—especially when there are many answer options, where demanding perfect consistency would make it almost impossible to achieve a correct score by chance. By preventing an excessively low chance level, this method also ensures that subsequent normalization reflects realistic trends.

## E. Normalization in Concept-wise Comparison

Since different core knowledge abilities are grounded in diverse cognitive science prototypes, they entail distinct distributions of question formats (e.g., true/false or multiple choice with 2–4 options), resulting in varying levels of chance accuracy and inherent difficulty. Therefore, normalization is essential to ensure fair comparisons across these abilities and to more robustly demonstrate "core knowledge deficits." To achieve this, we normalize the accuracy for each ability by dividing by its corresponding chance-level accuracy: $\text{acc}^i_{\text{norm}} = \frac{\text{acc}^i}{c^i}$, where $\text{acc}^i$ denotes the model's accuracy on the $i$-th core knowledge ability, and $c^i$ represents the chance-level probability for ability $i$. As shown in Fig. 13, we report both the raw and normalized accuracies (see Sec.4.1). Both measures exhibit a similar upward trend, further supporting our first finding of core knowledge deficits.

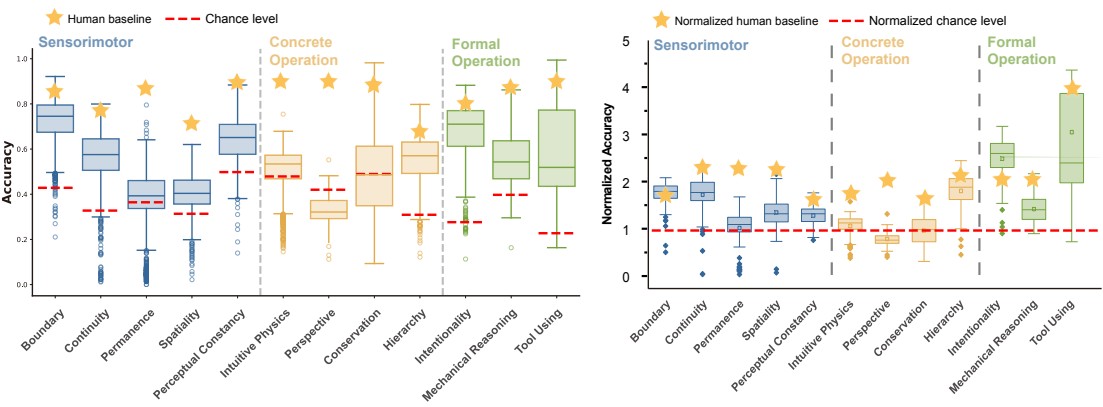

*Figure 13.* Accuracy by concept without normalization

## F. Performance with Different Prompts

*Table 5.* Illustration of 10 different prompts, in five categories.

| Category | Prompt |
|---|---|
| **no prompt** | [Empty String] |
| **think deep** | Let's think step by step. |
| | Take a deep breath and answer this question carefully. |
| **explanation** | Please answer the question and provide an explanation. |
| | Please answer the question and explain to me in simple terms. |
| | Please answer the question and explain it to me like I am 11 years old. |
| **reward & penalty** | Please answer the question carefully. I'm going to tip you 200 dollars for a better solution. |
| | Please answer the question carefully. You will be penalized if your answer is incorrect. |
| **bias mitigation** | Please answer the question and ensure that your answer is unbiased and doesn't rely on stereotypes. |
| **role playing** | You are an expert on cognitive science and are familiar with [Concept name]. |
| **cognitive instruction** | Please read the concept explanation and then answer the related question. Concept: [concept description]. |

We investigate the influence of different prompting techniques on MLLM performance using our benchmark. As shown in Tab. 5, we evaluate five prompt categories comprising ten distinct variants. We find that most prompts fail to improve performance. Notably, the cognitive instruction prompt (p10)—a concise description of the conceptual targets assessed by the task—outperforms all others, increasing accuracy by over 6%. This improvement likely stems from the fact that

explicitly stating the relevant core knowledge enables the model to more efficiently extract information that is otherwise distributed across its internal representations (Chalmers, 1992).

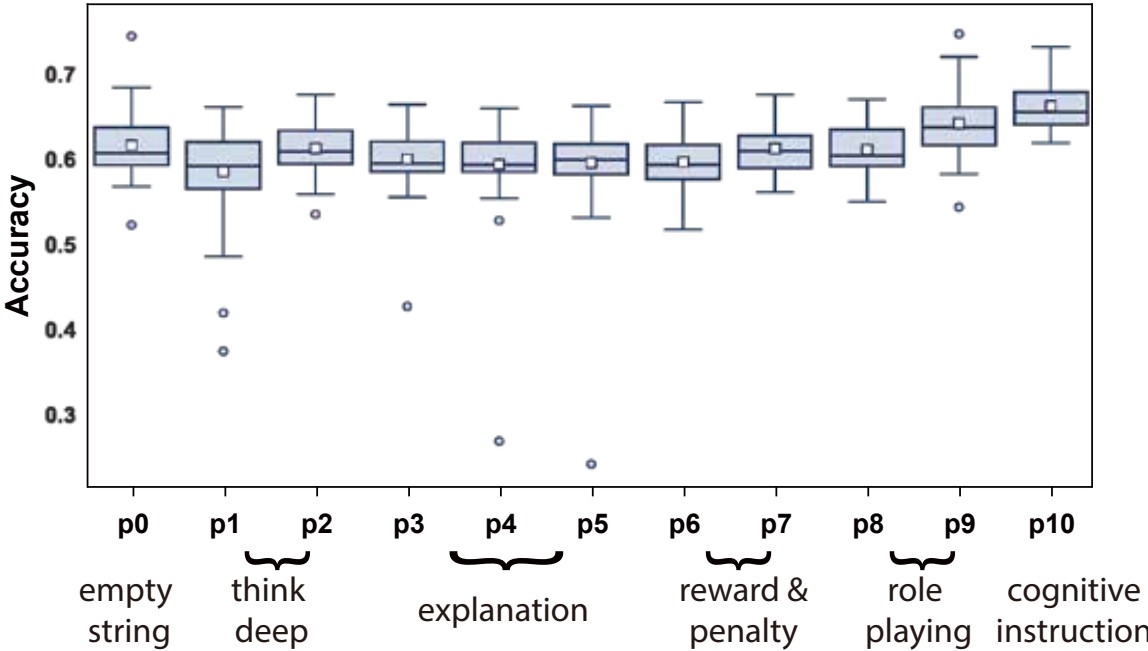

*Figure 14.* Performance of different prompts. Accuracy is averaged across all models.

This interpretation aligns with early insights from the connectionist literature, which suggest that distributed representations pose challenges for structured knowledge retrieval (Hinton et al., 1986; Chalmers, 1990). As networks scale, accessing specific conceptual structures becomes increasingly difficult—particularly for foundational concepts like core knowledge. Unlike high-level knowledge (e.g., historical events), which may be encoded in localized or clustered patterns, core knowledge tends to be diffusely represented across parameters due to its recurrence in diverse training contexts. This diffuse encoding makes such knowledge harder to isolate and systematically deploy in reasoning tasks. We hypothesize that cognitive instruction may servesas a retrieval cue, guiding the model's internal attention toward latent knowledge. However, we do not view this as a permanent or scalable solution. In real-world scenarios, models are unlikely to receive such explicit conceptual scaffolding, limiting the practical utility of this approach. Nevertheless, the finding points to a promising research direction for improving model reasoning through targeted scaffolding or memory-augmented mechanisms.

## G. Details on Statistical Testing

To statistically validate *core knowledge deficits*, defined as performance differences across the three developmental stages, we conducted paired $t$-tests. The test statistic is given by

$$t = \frac{\bar{d}}{s_d/\sqrt{n}},$$

where $\bar{d}$ denotes the mean difference between paired observations, $s_d$ is the standard deviation of these differences, and $n$ is the number of pairs.

Table 6 reports the results of the statistical test.

The extremely small $p$-values ($p \ll 0.001$) indicate that the observed performance differences between the Formal Operational stage and the other two stages are statistically significant.

Table 6. Paired t-test result.

| Contrast | $t$-statistic | $p$-value |
|---|---|---|
| Formal Operational vs. Concrete Operational | 22.68 | $4.79 \times 10^{-48}$ |
| Formal Operational vs. Sensorimotor | 28.15 | $3.81 \times 10^{-53}$ |

## H. Core Knowledge Deficits Under Different Conditions

We further validate "core knowledge deficit" depicted in Fig. 5 under different conditions. Specifically, we validate whether our finding 1 generalizes across different model subgroups (by competence and input constraints) and prompt variations.

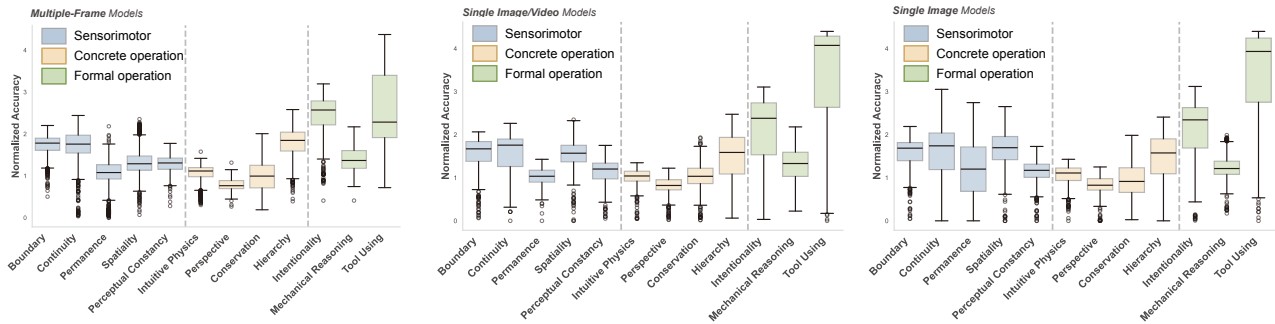

Figure 15. **Condition Group I: Model Input Constraints**

As shown in Fig.15, all three input formats and corresponding question subsets exhibit a pattern consistent with our conclusions in Sec.4.1.

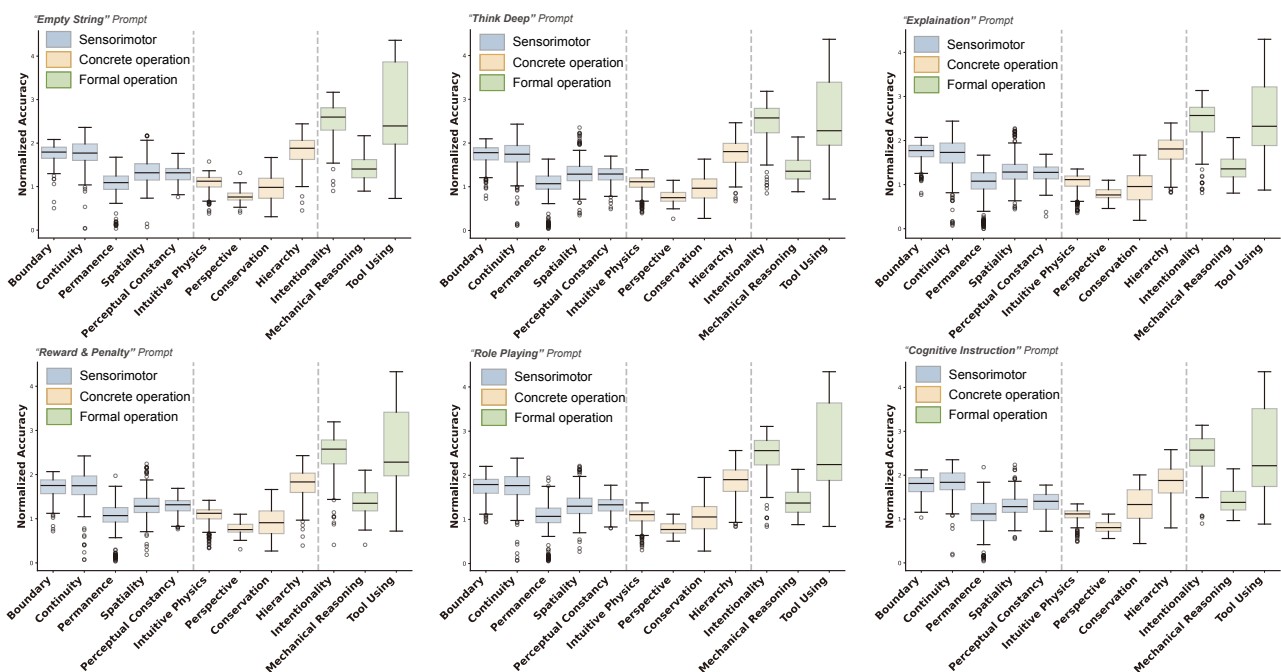

Figure 16. **Condition Group II: Prompt type**

Across Fig. 16, all six types of prompts yield patterns that align with our conclusions in Sec. 4.1.

As shown in Fig. 17, the overall performance pattern remains consistent across different model selections. This replicates

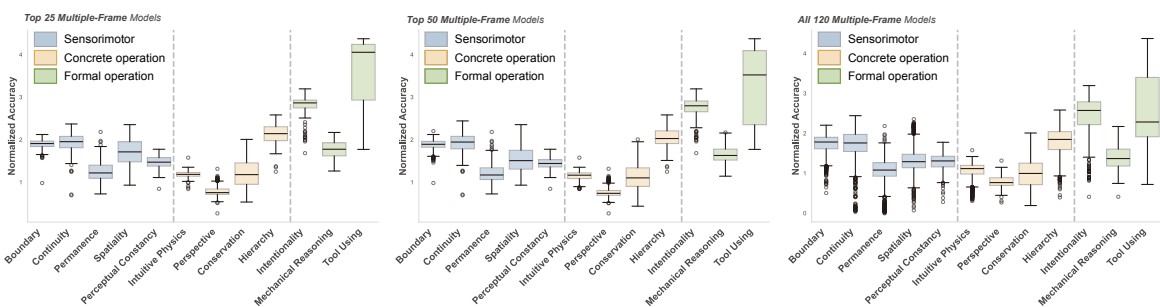

*Figure 17.* **Condition Group III: Model Competence. Top-n** refers to n models with the highest average accuracy out of 120 total models that have multiple-frame capacity and thus evaluated on all data of **CoreCognition**

the core knowledge deficit, with lower-level tasks consistently showing lower accuracy. Expanding the selection from the top 25 to all 120 multi-image models increases variability and slightly reduces median accuracy, particularly for higher-level tasks. This growing heterogeneity in model quality underscores that gains on abstract tasks are not uniformly reliable, and the the gap in core abilities persists.

# I. Dependencies Between Core Abilities Under Different Conditions

We further examine the stability of finding 2 depicted in Fig. 6 by validating it under different conditions.

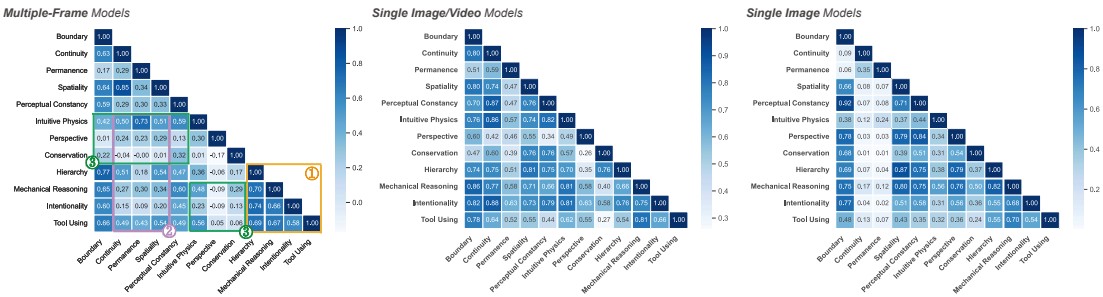

*Figure 18.* **Condition Group I: Model Input Constraints**

As shown in Fig. 18, the inter-concept dependencies in single-image and video model subgroups closely mirror those of the full-capacity multi-frame models discussed in Sec. 4.2. Two exceptions are Continuity and Permanence, which consistently exhibit low correlations with other concepts in the single-image setting. In particular, Permanence cannot be effectively assessed with single-image models, as this subset is unrepresentative and includes fewer than 10 questions.

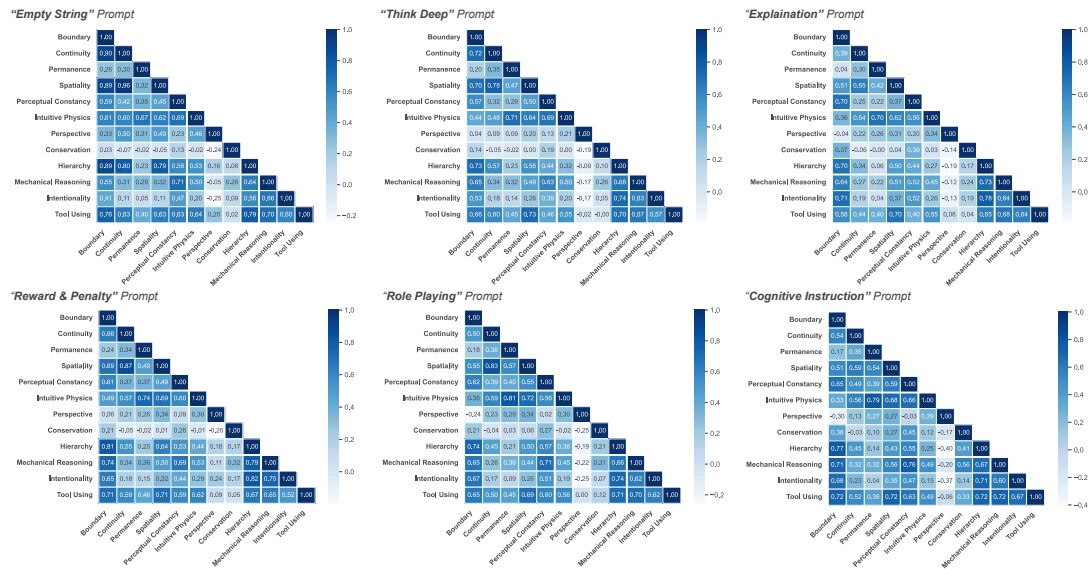

*Figure 19.* **Condition Group II: Prompt type**

As shown in Fig.19, all six prompts produce inter-concept dependency patterns consistent with our conclusions in Sec.4.2.

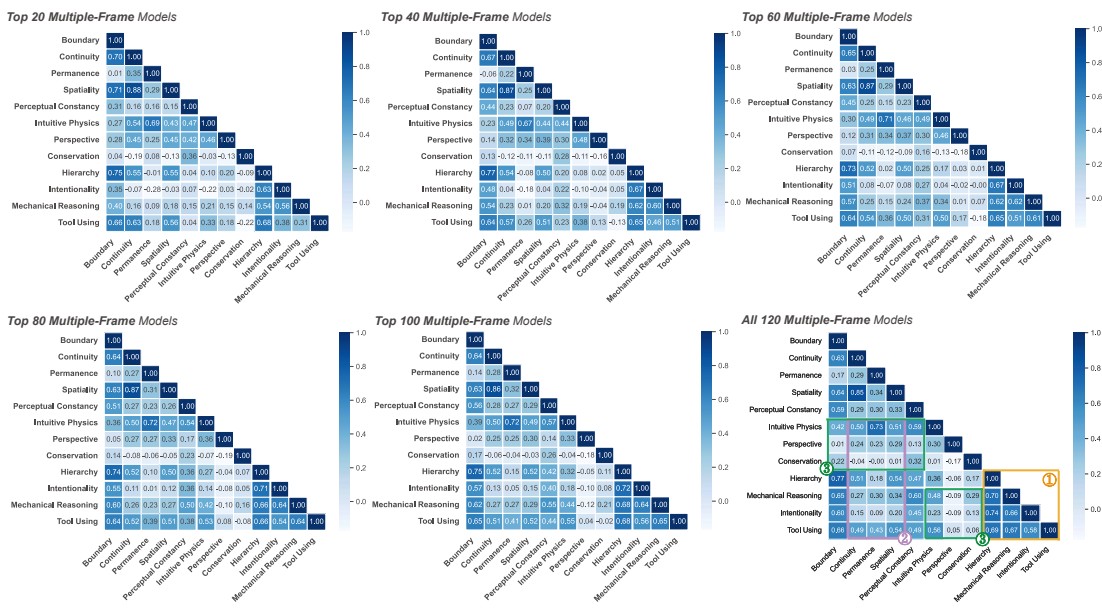

*Figure 20.* **Condition Group III: Model Competence. Top n** refers to n models with the highest average accuracy out of 120 total models that have multiple-frame capacity and thus evaluated on all data of **CoreCognition**

Across different subgroups in Figure 20, the overall pattern of inter-dependencies remains consistent with our original observation: while correlations between high-level abilities are significant, cross-stage correlations are generally modest, with continuity, Permanence, and Spatiality consistently show low correlations with higher-level abilities, while Perspective and Conservation remain largely uncorrelated with other tasks—reinforcing their isolation. Although a large number of models (e.g., Top 100, All 120) lead to slightly higher correlations, this likely reflects performance floor effects rather than genuine cognitive integration, as lower-performing models tend to fail relatively uniformly across tasks.

## J. Scaling Effect on Core Knowledge Under Different Conditions

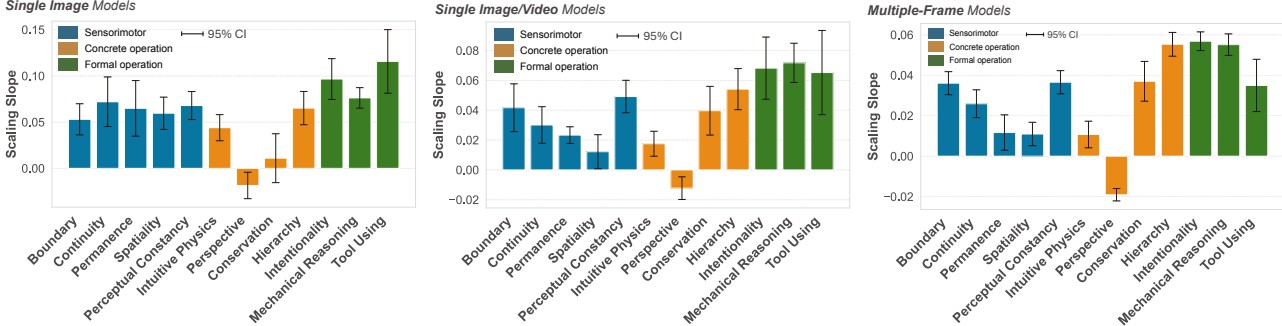

*Figure 21.* **Condition Group I: Model Input Constraints**

We also tested the scaling effect under different subgroups of models with different input constraints. As per Fig. 21, all three subgroups of models lead to a similar scaling effect aligned with our observations in Sec. 4.4. Further condition partitions (model competence and prompt type) are not analyzed for scaling effect because (1) it is the nature of the scaling analysis to study across model groups and (2) reducing the data size to a single prompt violates the data size requirement of a stable regression fit.

## K. Explanation of Concept Hacking with an Example

For example, as shown in the third case of Fig. 9, a standard probe of perceptual constancy assesses whether a model understands that a bridge of uniform width extending into the ocean does not actually become narrower in the distance. In the manipulated condition, all task-irrelevant details—such as the viewing angle and environmental textures—are kept identical to the standard task, but the bridge itself is altered to genuinely taper as it extends outward. Models possessing the understanding of perceptual constancy would have no difficulty answering both the manipulation task and standard control correctly. On the contrary, a model relying on spurious correlations between the task and previous examples of similar scenarios in the data would succeed in the original task but fail the manipulated one. Finally, a model with a strong inclination toward the belief that objects extending into the horizon are actually getting thinner physically would fail the control task while correctly answering the manipulated version due to its misaligned knowledge about the world.

# L. Qualitative examination of *Concept Hacking*

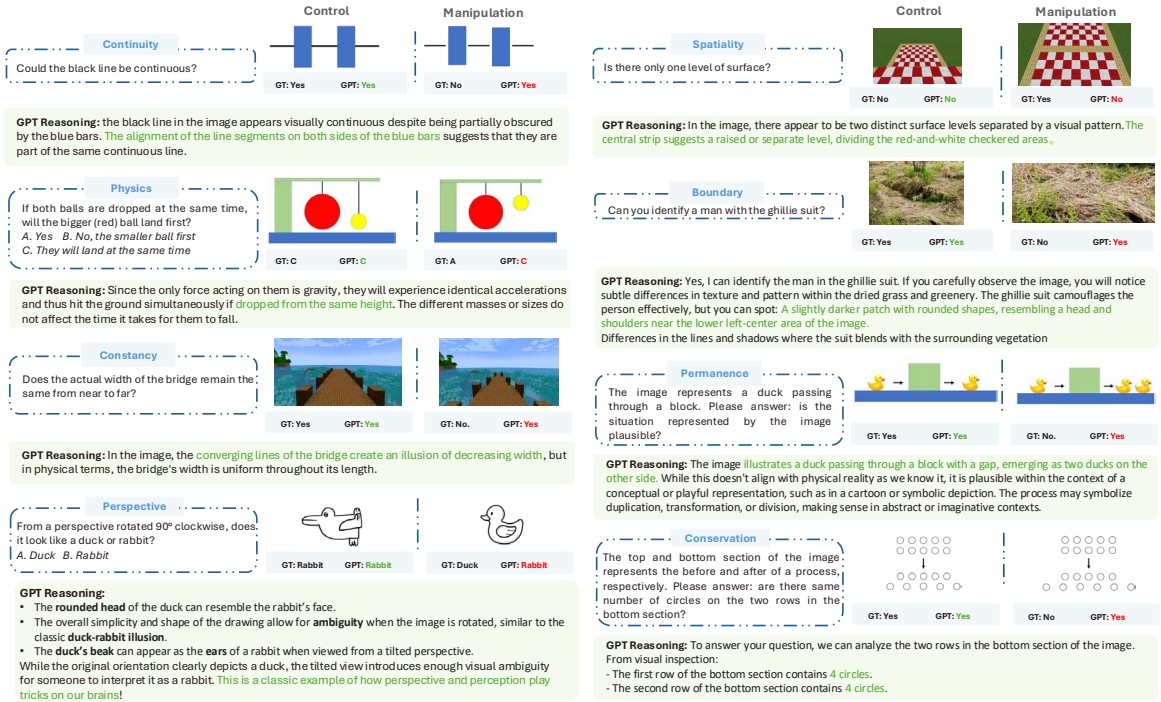

*Figure 22.* Example Questions from the Concept Hacking. Each example is presented with GPT-4o's answer (w/explanation) to the manipulated image.

We further conduct a qualitative examination of the reasons underlying the models' low performance on manipulation samples in Fig.22. Through prompting techniques such as Chain-of-Thought (Wei et al., 2022b) and "provide an explanation" (Li et al., 2022b), we are able to examine the reasoning chain behind the model's response. Using GPT-4o as a case study, we find that its low performance on manipulation samples primarily stems from an overreliance on shortcuts, possibly learned during training. When presented with manipulation samples, GPT-4 often reproduces reasoning similar to that used for corresponding control samples. For instance, in 22, GPT-4o correctly explains the illusion of decreasing bridge width (e.g., "the converging lines of the bridge create an illusion of decreasing width") even when the bridge's width was actually manipulated to decrease. This indicates that the model's reasoning was not grounded in the visual information explicitly presented in the image, but resorted to the shortcut understanding of the "near-large, far-small" phenomenon learned during the pretraining.

# M. Limitations

We acknowledge several limitations in our study. To begin with, VQA used in **CoreCognition** introduces auxiliary demands such as linguistic capabilities, counting, and object recognition. While we mitigated some through meticulous filtering of questions and rigorous validation across 11 distinct prompts, completely eliminating these auxiliary factors remains challenging. The VQA format also inherently restricts **CoreCognition**'s applicability to models capable of linguistic processing, narrowing the range of testable models. A complementary evaluation format is planned in future work to include non-linguistic or purely visual models. In addition, despite careful dataset design, MLLMs still exploit shortcuts and spurious features in **CoreCognition**, potentially inflating their performance, as evidenced in Sec. 5. Although *Concept Hacking* partially addresses this, its meticulous nature limits scalability, restricting its applicability to large-scale benchmarking.

