# OpenReview forum: "Core Knowledge Deficits in Multi-Modal Language Models"
_ICML.cc/2025/Conference — ICML 2025 poster_

### Official Review · Reviewer_nvFw · 2025-02-19

**Overall Recommendation:** 4

**Summary:**

This paper introduces the CoreCognition dataset focusing on a systematicity evaluation of multimodal models. Tasks in the benchmark are designed based on the well-established core-knowledge theory in developmental psychology. These tasks cover multiple aspects of human multimodal cognition, spanning from low-level perception tasks to high-level tasks that require tool use and strong commonsense knowledge. Evaluation of a large set of multimodal language models and correlation analysis between tasks & other benchmarks also justifies the contributions of the proposed benchmark.

**Claims And Evidence:**

Claims made in the paper are well supported by both empirical and computational evidence. Overall, I like how each tasks are supported by the corresponding developmental psychology background. Model evaluations were also thoroughly performed, I can see most state-of-the-art vision-language models are covered. Correlation analyses and ablation studies are good. I do think there is a flaw underlying the task design, as all the tasks require language---usually, language is treated as a separate aspect of core knowledge, and it might account for explaining why this paper found low-level tasks are harder for models than high-level tasks.

**Essential References Not Discussed:**

Some related benchmarks that also focus on evaluating core knowledge are missing in the reference, e.g., AGENT [A], and Binz & Schulz, 2023.

[A]. Shu, T., Bhandwaldar, A., Gan, C., Smith, K., Liu, S., Gutfreund, D., ... & Ullman, T. (2021, July). Agent: A benchmark for core psychological reasoning. In International conference on machine learning (pp. 9614-9625). PMLR.

[B]. Binz, Marcel, and Eric Schulz. "Using cognitive psychology to understand GPT-3." Proceedings of the National Academy of Sciences 120.6 (2023): e2218523120.

**Experimental Designs Or Analyses:**

Experimental designs and analyses reported in this paper sound good to me. One thing that this paper needs further discussion on is how the authors performed the human study, for example, how many human participants were recruited and how were they paid?

**Methods And Evaluation Criteria:**

Yes, the proposed evaluation criteria generally make sense. I do see one potential problem is the difficulties of all the tasks. In the developmental psychology literature, core knowledge usually focuses on children's ability to perform a basic understanding of the world, while the proposed tasks in this paper seem to be much harder problems that children cannot easily solve. However, this might be a minor issue as the authors are not considering studying the developmental or learning trajectory of these models.

**Other Comments Or Suggestions:**

N/A.

**Other Strengths And Weaknesses:**

Overall I like this paper due to i). the importance of introducing developmental insights to model evaluations, and specifically I think the core knowledge approach, as demonstrated in this paper, did a neat job. ii). model evaluations were carefully performed and I can clearly see some insights that might be helpful for the broader community.

Some potential weaknesses from this paper might include i). the VQA format, especially the fact that almost every task is entangled with language understanding, makes the core knowledge evaluated here not as clean as controlled experiments performed in the original developmental psych literature. This may lead to not directly comparable results (e.g., why low-level tasks are harder than high-level tasks for machines).

**Questions For Authors:**

How will the proposed dataset/benchmark be released and how will the benchmark be maintained? I recommend doing something like a dataset card as the common practice in neurips benchmark tracks.

**Relation To Broader Scientific Literature:**

This paper seems interesting to a broader audience, for example I think cogsci and developmental psychology people would be interested in the proposed benchmark.

**Theoretical Claims:**

There are no foreseeable issues with the theoretical claims made in this paper, as it is not a theory paper.

---

> ### Author Rebuttal · Authors · 2025-04-01
>
> ```>>> Q1``` The tasks are much harder than children cannot easily solve compared to developmental CogSci
> ```>>> A1``` Thanks for bringing up this nuance. We address the concern in two aspects. First, while all our tasks are derived from standard developmental cognitive science prototypes, e.g., "Three-Mountain", we extend the questions beyond the few typical examples from the textbooks to comprehensively evaluate the capabilities of MLLMs. This leads to a significantly larger and more systematic set of tasks and evaluations, rather than relying on the limited examples typically found in conventional literature. Second, it’s important to note that the subjects being evaluated are not infants or young children with limited acquired knowledge, but rather MLLMs adapted from large language models (LLMs), which are reported to possess human PhD-level knowledge and reasoning abilities [1].
>
> [1] OpenAI (2024). Learning to reason with LLMs. https://openai.com/index/learning-to-reason-with-llms/
>
> ---
>
> ```>>> Q2``` One thing that this paper needs further discussion on is how the authors performed the human study, for example, how many human participants were recruited and how were they paid?
> ```>>> A2``` We recruited college students as participants, each receiving the same questions as the MLLMS. To ensure attentiveness, we included 1% mismatched question-answer or question-image pairs, asking participants to flag unclear or complex items within 90 seconds. Participants were paid only if they passed the attention check. 22 participants' answers were accepted for final statistics.
>
> ---
>
> ```>>> Q3``` Providing more task examples in the supplementary would be more helpful.
> ```>>> A3``` Thanks for the advice. We have provided test cases in both the main paper and the supplementary. We will add more examples following your suggestion.
>
> ---
>
> ```>>> Q4``` Suggested references.
> ```>>> A4``` Thanks for the advice. We will include all suggested papers in our citations and provide an extended discussion about them.
>
> ---
>
> ```>>> Q5``` Language is treated as a separate aspect of core knowledge, and it might account for explaining why this paper found low-level tasks are harder for models than high-level tasks. The VQA format, especially the fact that almost every task is entangled with language understanding, makes the core knowledge evaluated here not as clean as controlled experiments performed in the original developmental psych literature. This may lead to not directly comparable results (e.g., why low-level tasks are harder than high-level tasks for machines).
> ```>>> A5``` Thanks for raising this concern. We will address your concern in the following two aspects.  First, indeed, we acknowledge that the VQA format is entangled with language understanding, as demonstrated by the impressive capabilities of large language models (LLMs) in both language comprehension and their use as a tool for reasoning [1, 2, 3], we argue that the influence of language bias is relatively minimal when evaluating core knowledge.
>
> Second, we contend that the core challenge lies in developing a scientifically robust evaluation methodology that can specifically probe the abilities of LLMs and MLLMs. Regardless of the evaluation method—whether VQA, retrieval tasks, or using output logits—assessing the model's specific capabilities requires auxiliary task demands [4]. In this context, the interaction between core knowledge and other factors (whether linguistic or otherwise) is inevitable, as LLMs inherently rely on language as a vehicle for expressing and processing information.
>
> We hope this clarifies our rationale for using the VQA format. We will discuss this shortcoming in the limitations Section.
>
> [1] Millière, Raphaël. "Language Models as Models of Language." arXiv, 2024, arXiv:2408.07144.
> [2] Brown, Tom, et al. "Language models are few-shot learners." Advances in neural information processing systems 33 (2020): 1877-1901.
> [3] Wei, Jason, et al. “Chain-of-Thought Prompting Elicits Reasoning in Large Language Models.” Proceedings of the 36th International Conference on Neural Information Processing Systems, NIPS '22, 2022.
> [4] Hu, Jennifer, and Michael C. Frank. "Auxiliary task demands mask the capabilities of smaller language models." arXiv, 2024, arXiv:2404.02418.
>
> ---
>
> ```>>> Q6``` How will the proposed dataset/benchmark be released, and how will the benchmark be maintained? I recommend doing something like a dataset card as the common practice in neurips benchmark tracks.
> ```>>> A6``` All data will be released upon acceptance and after internal review. Following standard procedures, we will open-source the dataset on GitHub and Hugging Face, including dataset cards, descriptions, and a viewer. Additionally, all model predictions and results will be made publicly available to ensure reproducibility.

---

### Official Review · Reviewer_Kcuh · 2025-03-06

**Overall Recommendation:** 3

**Summary:**

The paper investigates the hypothesis that the limitations of MLLMs in performing intuitive tasks, which are simple for humans, stem from the absence of "core knowledge"—innate cognitive abilities present from early childhood in humans. To explore this, the authors develop a novel benchmark called the CoreCognition dataset, designed specifically to assess these core cognitive concepts in MLLMs. The dataset covers 12 fundamental cognitive abilities and is used to evaluate 219 models across 10 different prompts, generating a total of 2,409 data points. The findings reveal that while MLLMs perform comparably to humans on complex, high-level cognitive tasks, they significantly underperform on simpler, low-level tasks. The study shows that there is no improvement in low-level cognitive abilities as model size and complexity increase, a stark contrast to high-level abilities which show scalable improvements. The authors also introduce “Concept Hacking” as an evaluation technique to demonstrate that MLLMs rely on superficial pattern recognition and shortcut learning rather than developing a genuine understanding of core knowledge.

**Claims And Evidence:**

Yes.

**Essential References Not Discussed:**

N/A. There are some similar works that tackles the core knowledge in evaluating LLMs but not MLLMs.

**Experimental Designs Or Analyses:**

Yes.

**Methods And Evaluation Criteria:**

Yes.

**Other Comments Or Suggestions:**

typo: line 029 double dots.

**Other Strengths And Weaknesses:**

From my point of view, I think my concerns mainly lines in the motivation behind the proposed benchmark. As mentioned, the paper’s motivation assumes that replicating human-like core knowledge is essential for the effective functioning of AI systems. This assumption is controversial and may not necessarily hold, as AI could potentially achieve high functionality through alternative means that do not mimic human cognitive processes. The debate on whether AI should replicate human cognition or develop its own unique methods remains unresolved and is a significant conceptual limitation of the study. Also, the motivation assumes that core knowledge can be clearly defined and operationalized within the context of AI systems, but from pure evaluation based on QA answers, this could lead to benchmarks that do not accurately reflect the underlying theories or mechanisms of core knowledge. Can this benchmark provide valuable insights into the training or designing of  MLLMs? How the benchmark results can help researchers improve their models? While the importance of core knowledge is widely acknowledged, I believe a good benchmark should serve purposes beyond simple evaluations.

**Questions For Authors:**

Given the inherent challenges in learning core knowledge from merely memorizing training data, how do you believe the CoreCognition benchmark can help advance the development of MLLMs in this domain? Can this benchmark provide valuable insights into the training or development strategies that might enable LLMs to acquire better core knowledge capabilities?

**Relation To Broader Scientific Literature:**

- By assessing MLLMs against a dataset designed to probe these core cognitive concepts, the paper directly investigates whether state-of-the-art AI systems possess analogous foundational skills, which is crucial for developing AI that can genuinely simulate human-like intelligence.
- The findings of the study provide a modern illustration of Moravec’s Paradox, showing that while MLLMs excel in high-level cognitive tasks, they struggle with basic cognitive tasks that are effortless for humans. Generally, the paper adds empirical data to ongoing debates about the nature of intelligence and complexity in AI systems.
- By introducing “Concept Hacking,” the paper adds a novel method to the repertoire of AI evaluation, specifically designed to test whether models truly understand the tasks they perform or merely capitalize on patterns and shortcuts.

**Theoretical Claims:**

N/A.

---

> ### Author Rebuttal · Authors · 2025-04-01
>
> ```>>> Q1``` typo: double dots.
> ```>>> A1``` Thanks. We will revise and remove all typos.
>
> ---
>
> ```>>> Q2``` Assume that replicating human-like core knowledge is essential for the effective functioning of AI systems is controversial and may not necessarily hold?
> ```>>> A2``` Thank you for the question! We approach this from 2 complementary perspectives: 1) The question presumes that core knowledge is unique to humans or human-aligned AGI. While this is resonable, it's also possible that core knowledge reflects more general cognitive principles or serves as mechanistic prerequisite that may be useful—or even necessary—for any form of general intelligence; such as chimpanzees showing behaviors aligned with core knowledge (Regolin and Vallortigara,1995; Santos, 2004; Lake 2017). This suggests that certain cognitive abilities emerge across intelligent systems, regardless of the path. Then even non-human pathways to AGI, e.g. scaling, might still lead to the emergence of these core abilities. In this light, a benchmark on core knowledge, such as ours, could offer a useful lens for evaluating the progress toward AGI, regardless of the path taken.
> 2) Even if core knowledge is a human-specific trait, the human trajectory remains a valuable source of insight given the current absence of clearly defined path to AGI. Despite great advancements, MLLMs continue to struggle with hallucination, lack of OOD generalization and robustness, indicating that important cognitive capacities may still be missing. Humans as the only model of high-level intelligence can serve as a useful reference for evaluating artificial systems. Moreover, if AGI were to eventually follow a human-aligned path, our benchmark would become vital in assessing high-level reasoning and perception grounded in foundational cognitive structures rather than driven by spurious shortcuts.
>
> ---
>
> ```>>> Q3``` Assumption that core knowledge can be clearly defined and operationalized within the context of AI systems, but from pure evaluation based on QA answers, this could lead to benchmarks that do not accurately reflect the underlying theories or mechanisms of core knowledge.
> ```>>> A3``` While isolating core knowledge dimensions is notoriously difficult, particularly in AI, there are certain aspects that makes this effort feasible. Extensive literature suggests that core knowledge exists and can be robustly probed and evaluated in humans, including infants who don't yet possess language or the ability to speak, as in Appendix B. The taxonomy of 12 core-abilities addresses this challenge by operationalizing core knowledge through a set of lower-level abilities. These abilities serve as tractable proxies for otherwise abstract cognitive dimensions and also connect naturally to stage-based developmental theories (Piaget, 1976; Rochat, 2024). This makes it possible to evaluate how models handle fundamental cognitive operations and how such abilities may scaffold more complex reasoning.
>
> ---
>
> ```>>> Q4``` From pure evaluation based on QA answers, this could lead to benchmarks that do not accurately reflect the underlying theories or mechanisms of core knowledge.
> ```>>> A4``` We acknowledge that the VQA format is intertwined with confounding factors, such as language understanding (see Q5 of Reviewer nvFw). However, evaluating specific abilities in AI models inherently requires auxiliary task demands (Hu and Frank, 2024)—regardless VQA, retrieval, or output logits—and every evaluations and benchmarks face more or less a similar challenge. To mitigate biases introduced by the VQA format, our effort includes
> - Instruct annotators to minimize the interplay of different abilities and exclude questions that require multiple competencies to answer.
> - Manually filter out ambiguous or confusing questions and use LLMs to enhance clarity and precision.
> - Extensive experiments with various phrasing and prompting techniques to alleviate biases arising from specific wordings or prompt formulations.
>
> ---
>
> ```>>> Q5``` Can this benchmark provide valuable insights into the training or designing of MLLMs? How the benchmark results can help researchers improve their models?
> ```>>> A5``` Our benchmark reveals that current training methods fail to effectively emerge these great abilities, suggesting future research into large-scale pretraining methods that can better scale core knowledge. More concretely, if core knowledge cannot be directly scaled, it may be beneficial to first teach or distill core knowledge into MLLMs prior to large-scale pretraining, enabling more data-efficient generalization akin to human learning. From an evaluation and design perspective, our benchmark uncovers distinct failure modes, including the lack of permanence, spatiality, boundary, and continuity. These limitations further hinder abilities such as visual perspective-taking and contribute to an over-reliance on shortcuts, which is a fundamental cause of poor out-of-distribution generalization.

---

### Official Review · Reviewer_vNKX · 2025-03-13

**Overall Recommendation:** 4

**Summary:**

The paper investigates core cognitive abilities in multimodal large language models. The authors find that models underperform in abilities that develop early in humans, while they perform comparable to humans on higher level abilities. They show that multimodal language models often rely on shortcut learning.

## update after rebuttal: I still think this is a solid paper and recommend acceptance.

**Claims And Evidence:**

- In the conclusion, the authors write "(2) models’ performance on high-level abilities does not correlate with the corresponding low-level abilities that ground them in humans". However, as shown in Figure 4 there are some correlations between lower level abilities and higher level abilities (boundary and continuity correlate quite well with intentionality and mechanical reasoning). While I agree in principle that higher correlations between low and high level abilities could be expected, this claim reads to strong to me.
- Furthermore, they write " (3) such abilities exhibit very low scalability among models, meaning that simply raising the number of parameters could not better the models’ performance on these abilities". While I agree that scale does not seem to yield perfect core abilities, it does look like almost every ability (except for perspective) benefits from scale in section 4.5. Sure, conservation and spatiality only slightly improve but all others seem to show some improvement with scale. Maybe this section would benefit from some numbers to make the plot a bit more digestible, such as "sensorimotor ability accuracy improves by 0.X on average". The resulting claim that "while scaling improves high-level reasoning, its effect on low-level cognitive abilities is minimal and, in some cases, even detrimental" therefore seems not perfectly supported, especially as the detrimental case (perspective taking) seems to have poor external validity.

**Essential References Not Discussed:**

There are three other works that are not mentioned and that showcase the weakness of VLMs on cognitive tasks. [1] investigates higher level visual cognition including intuitive physics. [2] also investigates intuitive physics in VLMs. [3] investigates visual illusions in VLMs and uses a technique that is similar to the "concept-based hacking" proposed here.

[1] Schulze Buschoff, Luca M., et al. "Visual cognition in multimodal large language models." _Nature Machine Intelligence_ (2025): 1-11.
[2] Balazadeh, Vahid, et al. "Synthetic Vision: Training Vision-Language Models to Understand Physics." _arXiv preprint arXiv:2412.08619_ (2024).
[3] Ullman, Tomer. "The Illusion-Illusion: Vision Language Models See Illusions Where There are None." _arXiv preprint arXiv:2412.18613_ (2024).

**Experimental Designs Or Analyses:**

-

**Methods And Evaluation Criteria:**

The choice of experiments seems principled. Also, the study investigates a large number of models.

**Other Comments Or Suggestions:**

Typos
- Two dots in the abstract on line 29
- Package import calls on line 1351 of the Appendix
- Line 169 right column "as" missing between "matching" and "the"?

Text color
- The yellow and red text color on page 5 is very hard to read and it is distracting in general.

Phrasing

- Line 233 in the left column reads "[...] a clear upward trend can be identified as the concepts move from low-level to high-level. This can be concluded as MLLMs perform worse on lower-level abilities than on higher-level ones, or in other words, there exist core knowledge deficits in Multi-Modal Language Models." Maybe I misunderstand but I thought all 12 abilities were considered core abilities. Sure, if MLLMs struggle on the lower-level abilities there are core knowledge deficits but this sentence reads as if the difference between performance on low to high level abilities is what constitutes core knowledge deficits.
- In 5.1 the authors write "At the core of the control experiment lies a _novel_ technique termed concept-based hacking". A similar technique has already been proposed for the investigation of visual illusions in VLMs in [3].

**Other Strengths And Weaknesses:**

The paper is original in that it investigates specific low level core abilities in VLMs. The experiments are thorough and performed on a large number of models. The investigation is definitely timely.

**Questions For Authors:**

1. Could the authors maybe speculate on why the perspective tasks shows such a poor external validity and is the only task that does not scale with the number of parameters? It seems like something might be off here.
2. In section 5, an agent with core knowledge should get control and manipulation right. A shortcut learner should get only the control right. And a non-core knowledge agent should only get the manipulation right. Basically, the latter would be a model that learns the wrong intuitions about basic visual properties, if I understand correctly. Now, the results seem to show that a large number of models actually fall into this category. Could the authors also speculate on why these models seem to learn wrong visual intuitions?

**Relation To Broader Scientific Literature:**

This paper contributes to a larger literature on how VLMs struggle with basic visual processing. It goes some way towards understanding what the missing components are that prohibit VLMs from having human-level visual understanding.

**Theoretical Claims:**

Not applicable

---

> ### Author Rebuttal · Authors · 2025-04-01
>
> ```>>> Q1``` "high-level abilities do not correlate with the corresponding low-level abilities" is too strong
> ```>>> A1``` Thanks! In Sec 4.3, the correlations between lower- and higher-level abilities are generally below 0.4. This is considerably lower than what's commonly observed in humans, though the phrasing may be strong. We will revise it to better describe the observation.
>
> ```>>> Q2``` Scaling analysis: add numbers to make the plot more digestible. "low-level cognitive abilities is minimal...even detrimental" seems not perfectly supported.
> ```>>> A2``` Thanks for the advice. Our current efforts include 1) comparison of scalability (slope) in the bottom left of Fig. 6, and 2) $R^2$ values represented by the size of the blobs to the left of the ability names in the upper-left sub-fig of Fig. 6.  We will also add specific numbers in the text to enhance clarity as suggested. We acknowledge that most abilities show some scalability; however, the scaling effect on lower-level abilities is significantly smaller (half the value) than that of higher-level abilities, which supports our conclusion. We will revise the text to better reflect the observation. Please see A4 below for a discussion on the perspective-taking exception.
>
> ```>>> Q3``` Suggested citation. A similar technique...for visual illusions in VLMs in [3].
> ```>>> A3``` Thanks for bringing this up. While [3] concurrently introduces a related idea, there are significant differences between the two. We offer a comprehensive controlled methodology grounded in the evaluation of core knowledge. Specifically, we formally propose a systematic procedure to manipulate input samples (${X}$) by flipping concept-level labels ($Y$) through targeted changes to causal features (${S}$) while holding non-causal features (${B}$) constant (Formally, a true predictive dist factorize as $p(Y|X) = \int p(Y| S, B) p(S, B| X)$). Whereas [3] presents only 10 samples of visual illusion (valuable but largely anecdotal), not situated within a broader diagnostic or theoretical framework with clear motivation and research questions. In addition, we present Figure 8,  showing the development trend of current MLLMs (with respect to scaling) is not in the ideal direction but biased towards illusion or shortcut. This adds an interpretable, diagnostic dimension to our evaluation that is absent in [3].
>
> ```>>> Q4``` "Worse on lower-level abilities...there exist core knowledge deficits". Aren't all 12 abilities core abilities?
> ```>>> A4``` All 12 abilities in our benchmark are grounded in core knowledge dimensions. However, lower-level abilities are operational approximations of basic cognitive systems and are thus more directly aligned with the notion of "core knowledge", while higher-level abilities are more abstract or compositional cognitive tasks. The observed upward trend in performance does not imply that only lower-level abilities reflect core knowledge. Rather, it suggests that while models may perform better on higher-level tasks—potentially by pattern matching or spurious correlation—they often struggle with the more fundamental reasoning required for lower-level tasks. This gap is what we refer to as a core knowledge deficit: the failure to demonstrate a robust understanding of the foundational abilities that higher-level tasks presuppose.
>
> ```>>> Q5``` Why do perspective tasks show such poor external validity and do not scale?
> ```>>> A5``` Thanks for the question, which relates to a key finding here. The perspective-taking task in our benchmark is based on the "Three Mountains" experiment, a type of level-2 perspective-taking (Moll, 2010), which requires mental simulation—the ability to build an internal model of the world and reason from it (Johnson-Laird, 1982). Whether MLLMs possess such internal world models is debated (Mitchell, 2023; Goddu et al., 2024) and the lack of scalability in perspective suggests current models may not. Unlike other tasks, perspective-taking additionally demands constructing a spatial model, the absence of which could explain the unexpected downward trend we observe in performance as model size increases.
>
> ```>>> Q6``` A large number of models learn wrong visual intuitions?
> ```>>> A6``` Core-illusion refers to models' response driven by a natural perception of the world, i.e., a lack of core knowledge. Models that learn from statistical correlations in the data may fall short in acquiring core abilities, as MLLMs trained on vast, multimodal datasets are often biased by shortcut signals, producing answers that resemble advanced reasoning but lack the conceptual grounding that allows humans to apply knowledge flexibly and consistently across contexts (Mitchell, 2023). Due to limited space, kindly refer to Reviewer ZrNC's A2 for a technical explanation of why such a distinction can be rooted in the pretraining process.

---

### Official Review · Reviewer_ZrNC · 2025-03-17

**Overall Recommendation:** 4

**Summary:**

This paper presents an evaluation framework for assessing the image understanding capabilities of multimodal language models (MLLMs) from a lens of cognitive taxonomy of concepts of learning. Inspired by the cognitive science literature on visual concept learning, the authors present a “CoreCognition” benchmark, which has been curated by defining a set of abstract learning ‘milestones’ or ‘skill levels’ of human visual cognition, and creating questions in a stratified manner per ‘skill level’ to controllably assess the image understanding capabilities of SOTA LLMs, bringing in a cognitive perspective into the benchmark curation. They ensure quality by conducting manual reviews of every question, ensuring that the options are cycled to mitigate positional bias in LLM answering, and testing various models under multiple prompted settings to regularize any instruction following effects.
By probing LLMs, the authors conclude that LLMs perform better on ‘higher level skills’ as opposed to more fundamental, ‘early developed’ core capabilities under their proposed taxonomy. Further, they test scaling of models to understand the trends of curriculum of tasks v/s model sizes, showing inconsistent behaviour e.g. more ‘core’ capabilities not scaling as expected. Finally, the authors throw light on ‘Concept Hacking’, highlighting the tendency of LLMs to rely on spurious correlations to solve questions of these kinds.

**Claims And Evidence:**

All the claims in the submission are supported by evidence where necessary. Specifically,
1. The claim on 'MLLMs consistently perform worse on low-level abilities compared to high-level abilities' is empirically supported by the evaluation of LLMs under these categories across different LLMs and prompting techniques.  The proposed benchmark has been curated by attempting to define a taxonomy which is in-line with the existing cognitive science literature, which is appreciated (and much needed) as we move towards an era of natively multimodal LLMs.  The taxonomy and abstraction of the proposed curriculum is a very first attempt and I'd be interested in understanding on why this taxonomy was chosen, and if there are any specific inspirations even from the ML literature e.g. OOD robustness, robotics.

2. The claim on "No observable scaling on low-level abilities with respect to increasing model parameters" is an interesting one: while I do understand empirical backing, did the authors expect otherwise and why? Do we _want_ models to follow the similar curriculum as humans do?

3. On control hacking, the claim that models rely on spurious correlations - this is not directly visible from Figure 8, and it would be greatly appreciated if the authors can intuitively explain the difference in illusions and shortcuts.  Nevertheless, it is good to note that Humans perform in a superior fashion.

**Essential References Not Discussed:**

There is rich literature on OOD robustness (taking the classic example of Waterbirds v/s Landbirds), e.g. [1] which could be referred to. Spurious correlations in the field isnt new, and I'd be curious to know if the space of LLMs for spurious correlations is any different from the space of spurious correlations for ResNets/ViTs etc.

[1] Distributionally Robust Neural Networks for Group Shifts: On the Importance of Regularization for Worst-Case Generalization
https://arxiv.org/abs/1911.08731

**Experimental Designs Or Analyses:**

Yes. As mentioned above, they ensure quality by conducting manual reviews of every question, ensuring that the options are cycled to mitigate positional bias in LLM answering, and testing various models under multiple prompted settings to regularize any instruction following effects.

**Methods And Evaluation Criteria:**

Yes. The method itself is a benchmark.

**Other Comments Or Suggestions:**

1. There is an extra dot in the abstract on line number 29.
2. It would be nice to have an explanation for every quadrant in Figure 8.

**Other Strengths And Weaknesses:**

**Strengths**:
1. Overall, I like the motivation to build cognitively inspired benchmarks for LLMs because it is much needed to bring in interdisciplinary perspectives to evaluation. Further, the evaluation methodology in the paper is rigorous.
2. I really appreciate the taxonomy that has been proposed: not only does this present a good benchmark, it also inspires other researchers in the field to conduct stratified analysis / training on such kind of data to improve their models specifically for specific facets of tasks.

**Weakness**
1. Considering the overall premise of the paper: Given that we are building benchmarks inspired by human cognition - do we a) _expect_  and b) _desire_ that our models follow _the same curriculum_ as humans do? Will models follow the same path to AGI as we humans learn?
I'd love to hear from the authors on what exactly they want the LLM community to take away from this paper when thinking in meta-terms about the overall capabilities, and how we scale these capabilities.

**Questions For Authors:**

Echoing the points I have mentioned above as questions:

1. Considering the overall premise of the paper: Given that we are building benchmarks inspired by human cognition - do we a) expect and b) desire that our models follow the same curriculum as humans do? Will models follow the same path to AGI as we humans learn? I'd love to hear from the authors on what exactly they want the LLM community to take away from this paper when thinking in meta-terms about the overall capabilities, and how we scale these capabilities.

2. It is very interesting to note that the 'cognitive instruction' seems to perform superiorly. This implies that certain cognitive priming to the
models can help boost accuracy on the benchmarks. Do the authors have any comments?

3. Is the space of LLMs for spurious correlations any different from the space of spurious correlations for ResNets/ViTs etc.? Can this benchmark be used to evaluate existing vision models? If not, what makes this eval benchmark specific to LLMs?

**Relation To Broader Scientific Literature:**

This paper proposes a standard benchmarking methodology.
The piece on shortcut learning is related to previous work on OOD robustness, an example linked below.

**Theoretical Claims:**

N/A

---

> ### Author Rebuttal · Authors · 2025-04-01
>
> We thank the reviewer for the valuable feedback. However, due to limited space, we could only answer the most significant questions here. We look forward to discussing the rest (i.e., illusion vs. shortcut and 4 quadrants of Fig 8, suggested citations, etc) in the discussion phase!
>
> ```>>> Q1``` Why this taxonomy was chosen, and if there are any specific inspirations from the ML literature e.g. OOD robustness, robotics?
>
> ```>>> A1```  We chose core knowledge as the basis for our taxonomy because it offers a theoretically grounded and developmentally validated account of the foundational systems underlying human cognition (Spelke & Kinzler, 2007). These dimensions are widely seen as essential to general intelligence, making them a meaningful lens for evaluating AI capabilities (Carey & Gelman, 1993; Carey, 2009; Spelke, 2022). While not directly inspired by robotics or OOD robustness, core knowledge is highly relevant to both, as generalization, transfer, and embodied reasoning rely on intuitive world understanding—precisely what core knowledge aims to capture. Thus, we provide a complementary perspective that can inform evaluations of generalization, transfer, and robustness in AI.
>
> ```>>> Q2``` The claim on "No observable scaling on low-level abilities..." Did the authors expect otherwise and why?...Do we a) expect and b) desire that our models follow the same curriculum as humans do? Will models follow the same path to AGI as we humans learn?...thinking in meta-terms about the overall capabilities, and how we scale these capabilities.
>
> ```>>> A2 ``` First, we do not expect otherwise as we anticipate challenges for the emergence of core-abilities simply through large-scale pretraining of statistical occurrence. Another potential reason is that, compared to high-level details required for complex tasks, core knowledge used in simpler tasks is spread across diverse contexts and the vast parameter space of the network (Shani et al., 2023), thus harder to isolate and apply systematically, leading to inconsistent or surface-level reasoning.
>
> It's a great question whether AI should follow the path of humans. We elaborate on a discussion in A2 of Reviewer Kcuh (due to limited space), that our paper neither argues for nor against the necessity of mirroring humans in pursuit of AGI. Rather, core knowledge is introduced as an evaluation for MLLMs, and the core knowledge deficit hypothesis is proposed as an explanation for the observed brittleness of MLLMs. More broadly, we hypothesize that core knowledge may represent a general principle of intelligence--human or otherwise. The critical research question is not whether models should mimic human learning, but how core abilities can emerge through scaling with cognitive or human-inspired adaptation, or other measurements.
>
> ```>>> Q3``` 'Cognitive instruction' seems to perform superiorly. This implies that certain cognitive priming to the models can help boost accuracy on the benchmarks. Any comments?
>
> ```>>> A3``` For now, we don't have a definitive explanation, but we find this effect aligned with early insights from the connectionist literature, which suggests that distributed representations pose a challenge for structured knowledge retrieval. As networks scale, retrieving specific conceptual structures becomes increasingly difficult (Hinton et al., 1986; Chalmers, 1990), especially for core knowledge. Unlike high-level knowledge, e.g., historical events, which are likely encoded in clustered patterns, core knowledge is highly distributed across the model’s parameters, as they are in multiple instances in training data, making them hard to isolate and deploy systematically for reasoning tasks (Garrigan, 2008; Green, 2024). We hypothesize that cognitive instruction may act as a retrieval cue that helps direct the model's internal attention toward latent but relevant knowledge. However, we do not believe this constitutes a permanent or scalable solution. In real-world environments, models are unlikely to receive such explicit guidance, limiting the practical utility of this strategy. Nevertheless, this result may point toward a promising research direction for improving reasoning via targeted scaffolding or memory-based mechanisms.
>
> ```>>> Q4``` Is the space of LLMs for spurious correlations any different from the space of spurious correlations for ResNets/ViTs, etc.? Can this benchmark be used to evaluate existing vision models? If not, what makes this eval benchmark specific to LLMs?
> ```>>> A4``` The space is different for LLM and vision models as the former is in an abstracted and discrete token space while the latter is high-dim pixel space. Our benchmark cannot be directly applied to vision models, since the questions are in VQA format (i.e., question answering). However, adaptation can be made to convert it into retrieval / binary classification for vision models to test. However, this falls out of the scope of this paper, and we leave it to future efforts.

---

### Decision · Program_Chairs · 2025-05-01

**Decision:**

Accept (poster)

**Comment:**

## Summary

This paper investigates whether multi-modal language models (MLLMs) possess "core knowledge" - fundamental cognitive abilities that humans develop at a young age. The authors develop a novel benchmark called the CoreCognition dataset, comprising 12 core cognitive concepts derived from developmental psychology literature. Using this benchmark, they evaluate 219 models with 10 different prompts, generating 2,409 total data points. Their key findings are that (1) MLLMs perform well on high-level cognitive tasks but struggle with low-level core abilities, (2) there is little correlation between performance on high-level and low-level abilities, and (3) low-level abilities show minimal scaling improvements with increased model size. The authors also introduce "Concept Hacking" to demonstrate that MLLMs rely on superficial patterns rather than genuine understanding.

## Strengths

- The paper brings valuable insights from cognitive science to AI evaluation, creating a benchmark that systematically probes fundamental cognitive capabilities.
- The experimental methodology is rigorous, evaluating a large number of models across multiple prompting strategies.
- The findings about differential scaling of high vs. low-level abilities offer important insights for model development.
- The concept hacking technique provides a novel way to distinguish between real understanding and shortcut learning.
- The work is interdisciplinary and has potential impact beyond AI, particularly in cognitive science.

## Weaknesses

- The use of language-based VQA format introduces potential confounds when evaluating pure visual cognition.
- Some claims regarding correlations between abilities might be slightly overstated.
- The relationship between human cognitive development and desirable AI capabilities remains an open question.
- More details on human studies methodology would strengthen the comparative analysis.

## Discussion

All four reviewers were positive about the paper, with three giving accepts and one giving a weak accept. The reviewers appreciated the paper's interdisciplinary approach, comprehensive evaluation, and novel insights about model scaling with respect to cognitive abilities.

The authors provided thorough responses to all reviewer concerns, clarifying their methodology, moderating some of their claims, and addressing questions about the relationship between human cognitive development and AI capabilities. They acknowledged limitations in their VQA format but provided reasonable justification for their approach.

I have read the authors' message to the AC regarding the lack of further reviewer engagement after rebuttal. While I understand their desire for more interactive discussion, the reviewers all acknowledged reading the rebuttal and maintained their positive assessments. The consistent endorsement from reviewers, along with the authors' thorough responses, provides sufficient basis for a decision.

## Recommendation

I recommend acceptance of this paper. It makes a significant contribution to our understanding of MLLMs through a novel cognitive science lens, with important implications for future model development. The authors should incorporate their clarifications from the rebuttal into the final version, particularly regarding task difficulty, human evaluation methodology, and the limitations of the VQA format.